# QUERY-AWARE FLOW DIFFUSION FOR GRAPH-BASED RAG WITH RETRIEVAL GUARANTEES

**Zhuoping Zhou[π], Davoud Ataee Tarzanagh[π], Sima Didari[π], Wenjun Hu, Baruch Gutow, Oxana Verkholyak, Masoud Faraki, Heng Hao, Hankyu Moon, Seungjai Min**

Samsung SDS Research America, Mountain View, CA, USA

[π]Equal Contribution.

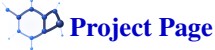 **Project Page**

## ABSTRACT

Graph-based Retrieval-Augmented Generation (RAG) systems leverage interconnected knowledge structures to capture complex relationships that flat retrieval struggles with, enabling multi-hop reasoning. Yet most existing graph-based methods suffer from (i) heuristic designs lacking theoretical guarantees for subgraph quality or relevance and/or (ii) the use of static exploration strategies that ignore the query's holistic meaning, retrieving neighborhoods or communities regardless of intent. We propose *Query-Aware Flow Diffusion RAG* (QAFD-RAG), a training-free framework that dynamically adapts graph traversal to each query's holistic semantics. The central innovation is *query-aware traversal*: during graph exploration, edges are dynamically weighted by how well their endpoints align with the query's embedding, guiding flow along semantically relevant paths while avoiding structurally connected but irrelevant regions. These query-specific reasoning subgraphs enable the first statistical guarantees for query-aware graph retrieval, showing that QAFD-RAG recovers relevant subgraphs with high probability under mild signal-to-noise conditions. The algorithm converges exponentially fast, with complexity scaling with the retrieved subgraph size rather than the full graph. Experiments on question answering and text-to-SQL tasks demonstrate consistent improvements over state-of-the-art graph-based RAG methods.

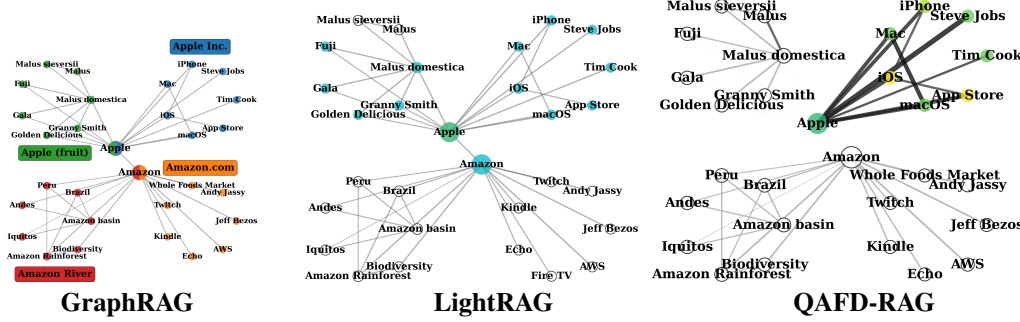

Figure 1: Comparison of graph-based RAG methods on Wikipedia pages (Apple fruit, Apple Inc., Amazon River, Amazon.com)[1]. Query: "Introduce Steve Jobs's products in Apple." **GraphRAG** (Edge et al., 2024b) retrieves entire communities, mixing relevant nodes (e.g., Mac, macOS) with irrelevant ones (e.g., Amazon River, Apple fruit). **LightRAG** (Guo et al., 2024) focuses on 1-hop neighborhoods, including both relevant nodes (e.g., Steve Jobs, iPhone) and structurally close but irrelevant ones (e.g., Fuji). **QAFD-RAG** reweights edges by the *query's holistic meaning*, suppressing irrelevant one-hop neighborhoods and preventing traversal into the Amazon River, Amazon.com, and Apple fruit clusters. The resulting subgraph is coherent, with edge thickness reflecting weight and node mass indicating importance (highest, lowest); see the discussion following Eqn. (5).

## 1  INTRODUCTION

Retrieval-Augmented Generation (RAG) enhances language models (LMs) by integrating external knowledge during generation (Fan et al., 2024; Gao et al., 2023b). A retriever first gathers relevant information for a query, which is then combined with the input and passed to the generator (Karpukhin et al., 2020). This is especially useful for question answering (QA) (Xiong et al., 2020; Zhu et al., 2021), where added context improves accuracy in domains such as healthcare, law, finance, and education (Xu et al., 2024; Zhang et al., 2023). With recent advances in large LMs (LLMs), RAG has become increasingly important for trustworthy AI, helping mitigate hallucinations (Tonmoy et al., 2024), improve transparency (Kim & Lee, 2024), adaptability (Shi et al., 2024; Wang et al., 2023), privacy (Zeng et al., 2024a;b), fairness (Shrestha et al., 2024), and reliability (Fang et al., 2024a).

Although conventional RAG is effective for unstructured document retrieval, real-world knowledge often has graph-structured forms—such as database schemas, social networks, and biomedical repositories. These structures preserve relational information that similarity-based retrieval misses, including multi-hop dependencies, hierarchies, and complex interactions. Graph-oriented RAG leverages such properties through techniques such as community detection and graph neural networks (Edge et al., 2024b; Wang et al., 2024), extending beyond similarity search to capture topological and semantic relationships (Fan et al., 2024; Gao et al., 2023b). This enables advanced multi-hop reasoning crucial for tasks such as text-to-SQL generation, scientific discovery, and medical diagnosis, where understanding relationships is as important as retrieving facts (Chen et al., 2025b; Wigh et al., 2022; Zhang et al., 2024c). For an overview, see the survey by Han et al. (2024).

However, existing graph-based RAG methods (Han et al., 2024) suffer from heuristic designs lacking theoretical guarantees for subgraph quality or relevance and/or the use of static exploration strategies that ignore the query's holistic meaning during traversal. As shown in Figure 1, GraphRAG (Edge et al., 2024b) applies uniform community detection regardless of query relevance, while LightRAG (Guo et al., 2024) extracts ego-networks around seed nodes without semantic alignment. In contrast, QAFD-RAG incorporates query semantics by reweighting edges and propagating mass along semantically aligned paths. Nodes on the reasoning chain (Apple → Mac → macOS) are emphasized with stronger colors, while irrelevant ones (Amazon River, Fuji) fade due to suppressed flow. This reweighting turns diffusion into a semantic filter, producing compact, interpretable subgraphs aligned with user intent. By contrast, holistic query-agnostic methods often include irrelevant nodes, omit distant but relevant information, and return unstructured lists rather than coherent reasoning paths. Lacking theoretical foundations, these heuristics also yield unpredictable performance.

Given these limitations, we ask:

> *Q: Under what conditions can we establish recovery guarantees for retrieving subgraphs that adapt to a query's holistic meaning in graph-based RAG?*

To address this, we turn to graph diffusion theory and in particular *flow diffusion*—the process of spreading mass from seed nodes to neighbors along graph edges (Lovász, 1993; Chung, 1997; Fortunato, 2010; Spielman & Teng, 2013b; Fountoulakis et al., 2023b). Spectral diffusion methods are effective for clustering and community detection due to strong guarantees and efficiency, but have not been studied in the presence of queries. We reformulate diffusion for graph-based RAG, linking flow/traversal to a query's holistic meaning. Specifically, we propose Query-Aware Flow Diffusion RAG (QAFD-RAG), a framework for dynamic, query-aware graph traversal via principled flow diffusion. The main contributions of this work are:

 **C1.  Query-Aware Flow Diffusion Framework:** We introduce the first principled flow diffusion method for graph-based RAG that incorporates query semantics via alignment-based edge weighting. QAFD-RAG adapts flow probabilities online, guiding traversal toward semantically relevant regions with complexity scaling with the retrieved subgraph size rather than the full graph (Figures 1 and 2).

 **C2.  Optimization and Statistical Guarantees:** We provide the first rigorous analysis of query-aware traversal with provable guarantees. Our results (Theorem 3) show exponential convergence to a unique query-dependent stationary distribution, and recovery guarantees (Theorem 7) ensuring relevant subgraphs are retrieved with high probability under mild signal-to-noise conditions.

 **C3.  Experimental Validation:** We evaluate QAFD-RAG on general QA (Tables 1 and 5), multi-hop QA (Table 3), long-document summarization (Table 2), and text-to-SQL (Table 4). Across these

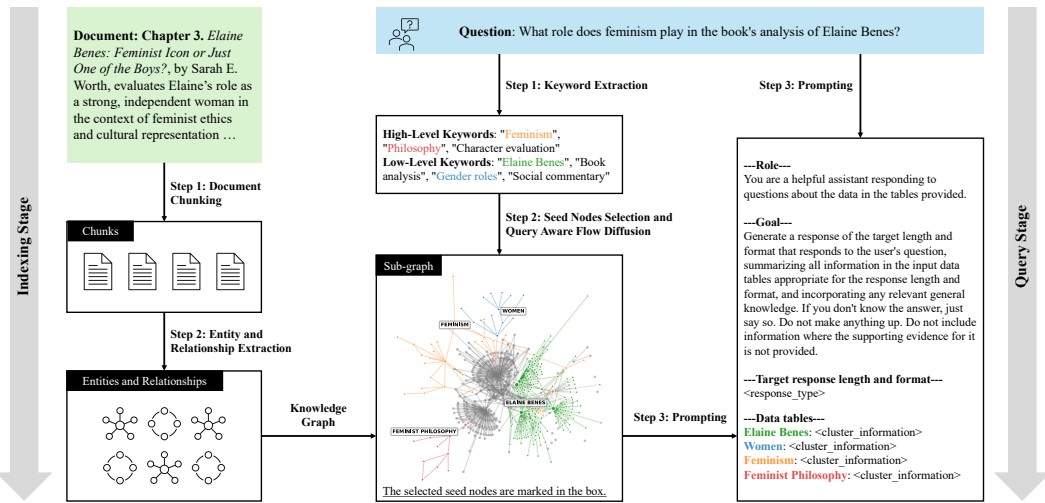

Figure 2: Two-stage QAFD-RAG framework: the indexing stage builds a KG from documents, and the query stage applies QAFD to extract and prompt subgraphs for response generation.

benchmarks, QAFD-RAG consistently outperforms state-of-the-art graph-based RAG baselines and improves over text-to-SQL baselines.

**Notation.** Let $\mathbb{R}^d$ be $d$-dimensional Euclidean space, and let $\mathbb{R}^d_+$ and $\mathbb{R}^d_{++}$ denote the nonnegative and strictly positive orthants. Vectors and matrices are in boldface (e.g., $\mathbf{a}$, $\mathbf{A}$) with entries $a_i$ and $a_{ij}$. For a matrix $\mathbf{A}$, $\mathbf{A}_{i:}$ and $\mathbf{A}_{:j}$ denote its $i$th row and $j$th column, and $\mathbf{A} \succ 0$ and $\mathbf{A} \succeq 0$ mean positive definite and positive semidefinite. For vectors, $\|\mathbf{a}\|_1 := \sum_i |a_i|$ and $\|\mathbf{a}\| := \left( \sum_i |a_i|^2 \right)^{1/2}$. For matrices, $\|\mathbf{A}\|_F := \left( \sum_{i,j} |a_{ij}|^2 \right)^{1/2}$. For $n \in \mathbb{N}$, define $[n] := \{1, \ldots, n\}$. We use standard asymptotic notations $O, \Omega, \Theta, o$, and $\omega$ (as $n \to \infty$). We write $\lhd$ for a fixed deterministic tie-breaking order on any finite set. For a finite set $U$ and scores $f : U \to \mathbb{R}$, $\texttt{Top-}k_{u \in U}(f(u))$ returns the $k$ elements of $U$ with largest scores, breaking ties by $\lhd$; equivalently, if $u_{(1)}, \ldots, u_{(|U|)}$ are sorted in decreasing order of $(f(u), \lhd)$, then $\texttt{Top-}k_{u \in U}(f(u)) := \{u_{(1)}, \ldots, u_{(k)}\}$.

## 2 METHODOLOGY

### 2.1 FRAMEWORK OVERVIEW

QAFD-RAG is a training-free, graph-based reasoning framework for RAG tasks. The framework operates in two phases: an *Indexing Stage* (IS), which builds a knowledge graph (KG) from raw documents (Chen et al., 2025a; Guo et al., 2024), and a *Query Stage* (QS), which introduces our key contribution—query-aware flow diffusion. This mechanism integrates query semantics into graph traversal, enabling adaptive retrieval with statistical guarantees. Unlike SOTA RAG methods such as GraphRAG (Edge et al., 2024b), which applies static community detection, or LightRAG (Guo et al., 2024), which extracts ego-nets (Figure 1), QAFD-RAG dynamically reweights edges and diffuses flow according to query alignment, suppressing irrelevant clusters and highlighting reasoning paths.

In IS, *IS-Step 1: Document Chunking* splits documents into context-preserving chunks; *IS-Step 2: Entity and Relationship Extraction* uses LLM prompting to build a structured KG. In QS, *QS-Step 1: Keyword Extraction* pulls conceptual and surface terms (e.g., *Feminism, Elaine Benes*) for broad coverage (Figure 2, Step 1). Next, *QS-Step 2: Seed Node Selection and QAFD* scores nodes by similarity to query keywords as detailed in Algorithm 1. For example, in Figure 1, *Steve Jobs* (0.95), *Apple* (0.92), and *iPhone* (0.88) are selected, while irrelevant ones (e.g., *Amazon*, 0.15) are excluded. Top-scoring nodes serve as seeds where mass is injected and propagated via flow diffusion (Figure 2, Step 2). Edges are dynamically reweighted to blend structure and semantics through the query-aware edge weighting mechanism described in Section 2.2, suppressing irrelevant expansions (Apple → Amazon River/fruit) and reinforcing meaningful ones. We formulate this diffusion process as a constrained optimization problem, solved efficiently via a push–relabel algorithm (Algorithm 2).

Recovery guarantees and algorithm complexity are provided in Section 3. Finally, *QS-Step 3: Response Generation* summarizes retrieved communities (e.g., *Elaine Benes*, *Feminism*) and prompts a downstream LLM, yielding grounded, consistent answers along reasoning paths.

## 2.2 QUERY-AWARE FLOW DIFFUSION

We cast subgraph retrieval as a QAFD problem. Given a natural language query $q \in \mathcal{Q}$ and a knowledge graph $\mathcal{G} = (\mathcal{V}, \mathcal{E}, \mathcal{R})$, with $\mathcal{V}$ entities, $\mathcal{E}$ edges, and $\mathcal{R}$ relations, the task is to retrieve a subgraph $\mathcal{G}_q \subseteq \mathcal{G}$ capturing reasoning paths relevant for downstream language model processing. We first recall flow diffusion from graph diffusion theory (Lovász, 1993; Chung, 1997; Spielman & Teng, 2013b; Fountoulakis et al., 2023b).

**Definition 1** (Flow Diffusion). *Flow diffusion spreads an initial amount of "mass" or "information" from seed nodes through a graph along its edges. At each step, mass is divided among neighbors by edge weights, so strongly connected nodes receive more and weakly connected ones less.*

Intuitively, flow diffusion resembles water or dye moving through pipes: most flow follows stronger pipes (high-weight edges), little through weaker ones. While well studied in clustering and community detection, it has not been explored in the presence of queries. We reformulate diffusion for graph-based RAG, linking traversal to query semantics. The first step is seed node selection, as seeds are usually assumed *a priori* in diffusion methods (Spielman & Teng, 2013b; Wang et al., 2017). Seed selection identifies query-relevant entities as starting points for flow diffusion. We adopt a semantic similarity–based approach and treat seed utility as approximately additive under an independence assumption; see Algorithm 1. Let $\mathcal{Q}$ be the query space and $\mathcal{W}$ the vocabulary space.

---

**Algorithm 1** Seed Node Selection

**Input:** Query $q$, graph $\mathcal{G} = (\mathcal{V}, \mathcal{E}, \mathcal{R})$ with node embeddings $\{e_v\}_{v \in \mathcal{V}}$, embedding function $h(\cdot)$, similarity $H_{\text{sim}}(\cdot, \cdot)$, number of seeds $N$
**Output:** Seed entities $\mathcal{S}_{\mathcal{V}} \subseteq \mathcal{V}$ with $|\mathcal{S}_{\mathcal{V}}| = N$
1: Extract keywords $\mathcal{K}_q = \{w_1, \ldots, w_{|\mathcal{K}_q|}\}$ from $q$ using an LLM
2: Compute $e_{q,i} = h(w_i)$ for all $i = 1, \ldots, |\mathcal{K}_q|$
3: **for all** $v \in \mathcal{V}$ **do**
4: $\quad$ $\text{score}(v, q) \leftarrow \max\limits_{i \in \{1, \ldots, |\mathcal{K}_q|\}} H_{\text{sim}}(e_{q,i}, e_v)$
5: **end for**
6: $\mathcal{S}_{\mathcal{V}} \leftarrow \text{Top-}N_{v \in \mathcal{V}}(\text{score}(v, q))$
7: **return** $\mathcal{S}_{\mathcal{V}}$

---

Define $g : \mathcal{Q} \to 2^{\mathcal{W}}$ mapping a query $q \in \mathcal{Q}$ to keywords $\mathcal{K}_q := \{w_1, \ldots, w_{|\mathcal{K}_q|}\}$ via LLM prompting (Appendices A4 and A5). Let $\mathcal{K}_{\mathcal{V}} := \{k_v : v \in \mathcal{V}\}$ be node identifiers. Let $h : \mathcal{W} \cup \mathcal{V} \to \mathbb{R}^d$ denote an embedding function that maps query keywords and graph nodes to $\mathbb{R}^d$:

$$\mathcal{E}_q := \{h(w_i)\}_{i=1}^{|\mathcal{K}_q|} = \{e_{q,i} \in \mathbb{R}^d\}_{i=1}^{|\mathcal{K}_q|}, \quad \text{and} \quad \mathcal{E}_{\mathcal{V}} := \{h(v)\}_{v \in \mathcal{V}} = \{e_v \in \mathbb{R}^d\}_{v \in \mathcal{V}}.$$

We then define a similarity function $H_{\text{sim}} : \mathbb{R}^d \times \mathbb{R}^d \to \mathbb{R}$ for semantic relatedness, with common choices including cosine similarity $e \cdot e' / \|e\| \|e'\|$, dot product $e \cdot e'$, or RBF kernel $\exp(-\gamma \|e - e'\|^2)$. The relevance score of a node $v$ for query $q$ is the maximum similarity across keywords via Step 4. We then select the $N$ highest-scoring entities via Step 6, breaking ties arbitrarily. For small graphs, keywords and seeds can often be extracted in one LLM prompt (Appendix A5). For large graphs (e.g., Table 11), we apply Algorithm 1, which runs in $\mathcal{O}(|\mathcal{V}| |\mathcal{K}_q| d + |\mathcal{V}| \log N)$.

**Dynamic Query-Aware Edge Weights.** Given seeds $\mathcal{S}_{\mathcal{V}}$ and query $q \in \mathcal{Q}$, we perform flow diffusion to extract reasoning subgraphs. Traditional diffusion uses static edge weights that ignore query context, leading to uniform exploration. Our key insight is to make edges *query-aware gates* that modulate flow strength using both structural connectivity and query alignment, enabling recovery guarantees. For each $q \in \mathcal{Q}$, edge $(u, v) \in \mathcal{E}$, and $a, b, c \geq 0$, the query-aware edge weights are

$$\bar{w}(q, u, v) := c \cdot H_{\text{sim}}(h(u), h(v)) \circ (a + b \cdot (H_{\text{sim}}(h(u), h(q)) \circ H_{\text{sim}}(h(v), h(q)))), \quad (1)$$

where $h(\cdot)$ is the embedding function, $H_{\text{sim}}$ a similarity measure, $\circ$ a binary operation (addition or multiplication), and $a, b, c \geq 0$ hyperparameters.

We explore three variants that balance structural and semantic signals differently:

$$\bar{w}_{\text{Mean}}(q, u, v) := \tfrac{1}{3}(H_{\text{sim}}(h(u), h(v)) + H_{\text{sim}}(h(u), h(q)) + H_{\text{sim}}(h(v), h(q))), \quad (2a)$$

$$\bar{w}_{\text{Product}}(q, u, v) := H_{\text{sim}}(h(u), h(v)) \cdot H_{\text{sim}}(h(u), h(q)) \cdot H_{\text{sim}}(h(v), h(q)), \quad (2b)$$

$$\bar{w}_{\text{Hybrid}}(q, u, v) := H_{\text{sim}}(h(u), h(v)) \cdot (a + b(H_{\text{sim}}(h(u), h(q)) + H_{\text{sim}}(h(v), h(q)))). \quad (2c)$$

Here, $H_{\text{sim}}(h(u), h(v))$ captures structural (node–node) similarity, while $H_{\text{sim}}(h(u), h(q)) \circ H_{\text{sim}}(h(v), h(q))$ measures query relevance to nodes $u$ and $v$. The multiplicative interaction in Product and Hybrid amplifies edges between query-relevant nodes while exponentially suppressing edges to irrelevant regions, transforming diffusion into a semantic filter. Weights are computed online: *when a query arrives, we first compute its embedding and then update only the edge weights* $\bar{w}(q, u, v)$ *encountered during traversal*. In experiments, $\bar{w}_{\text{Hybrid}}(q, u, v)$ yields slightly better results.

**QAFD's Primal–Dual Formulation.** Given approaches for seed nodes and dynamic weight computations $\bar{w}(q, u, v)$, we formulate flow diffusion (Definition 1) as a constrained optimization problem that minimizes total flow cost while enforcing mass conservation at each node. Let $\mathbf{f} \in \mathbb{R}^{|\mathcal{E}|}$ denote edge flows, $\bar{\mathbf{W}}(q) \in \mathbb{R}^{|\mathcal{E}| \times |\mathcal{E}|}$ the diagonal matrix of query-aware edge weights, and $\mathbf{B} \in \mathbb{R}^{|\mathcal{V}| \times |\mathcal{E}|}$ the incidence matrix of $\mathcal{G}$. The optimization is

$$\min_{\mathbf{f} \in \mathbb{R}^{|\mathcal{E}|}} \frac{1}{2} \mathbf{f}^\top \bar{\mathbf{W}}(q) \mathbf{f} \quad \text{subj. to} \quad \boldsymbol{\Delta} + \mathbf{B} \bar{\mathbf{W}}(q) \mathbf{f} \le \mathbf{T}, \qquad \forall q \in \mathcal{Q}. \tag{3}$$

Here, $\boldsymbol{\Delta} \in \mathbb{R}^{|\mathcal{V}|}$ encodes source mass injection and $\mathbf{T} \in \mathbb{R}^{|\mathcal{V}|}$ represents sink capacities. We note that in previous formulations (Wang et al., 2017; Fountoulakis et al., 2020b; 2023b), $\bar{\mathbf{W}}(q)$ is query-independent and, in most cases, is restricted to be the identity matrix. Following Definition 1, the sink capacity vector $\mathbf{T}$ determines how much mass each node can hold before diffusing to neighbors. A common choice is degree-based capacity where $T_v = \text{degree}(v)$, which allows high-degree nodes to accumulate more mass proportional to their connectivity, though uniform capacity $T_v = \beta$ for some constant $\beta > 0$ provides an alternative that treats all nodes equally regardless of their structural prominence. The source vector $\boldsymbol{\Delta}$ controls the initial mass distribution across nodes, with a common choice $\Delta_v = \alpha \sum_{u \in \mathcal{V}} T_u$ for source nodes $v \in \mathcal{S}_\mathcal{V}$ and $\Delta_v = 0$ otherwise, where $\alpha > 0$ controls source strength.

The corresponding Lagrangian of (3) with multiplier vector $\mathbf{x} \in \mathbb{R}_+^{|\mathcal{V}|}$ is

$$\mathcal{L}(\mathbf{f}, \mathbf{x}; q) = \tfrac{1}{2} \mathbf{f}^\top \bar{\mathbf{W}}(q) \mathbf{f} + \mathbf{x}^\top \left( \boldsymbol{\Delta} + \mathbf{B} \bar{\mathbf{W}}(q) \mathbf{f} - \mathbf{T} \right), \qquad \forall q \in \mathcal{Q}. \tag{4}$$

Taking $\partial \mathcal{L} / \partial \mathbf{f} = 0$ gives the primal-dual relation $\mathbf{f} = -\mathbf{B}^\top \mathbf{x}$. Substituting this into (4) and setting $\mathbf{L}(q) = \mathbf{B} \bar{\mathbf{W}}(q) \mathbf{B}^\top$ yields the dual problem

$$\min_{\mathbf{x} \in \mathbb{R}_+^{|\mathcal{V}|}} F(\mathbf{x}; q) := \tfrac{1}{2} \mathbf{x}^\top \mathbf{L}(q) \mathbf{x} + \mathbf{x}^\top (\mathbf{T} - \boldsymbol{\Delta}), \qquad \forall q \in \mathcal{Q}. \tag{5}$$

It is often more convenient to use (5), which incorporates the $\mathbf{T}$ and $\boldsymbol{\Delta}$ constraints into the objective. The solution provides node importance scores $\mathbf{x} \in \mathbb{R}_+^{|\mathcal{V}|}$, indicating each entity's query relevance. Intuitively, $\mathbf{f}$ represents edge flow strength—thicker edges in Figure 1(right)—while $\mathbf{x}$ denotes node mass or importance, shown by color in Figure 1(right): greener nodes are most relevant, yellow least, with intermediate shades indicating moderate importance.

Algorithm 2 performs coordinate-wise optimization updates on the dual objective (5) for a query and seed node. We set $\Delta_v = \alpha \sum_{u \in \mathcal{V}} T_u$ for $v \in \mathcal{S}_\mathcal{V}$ and $\Delta_v = 0$ otherwise. At each step, a node $v$ with excess mass ($m_v > T_v$) is chosen uniformly at random and $x_v$ is increased to reduce the objective. The mass vector $\mathbf{m}$ tracks the gradient of $F(\mathbf{x})$, initialized as $\mathbf{m} = \boldsymbol{\Delta}$ at $\mathbf{x} = \mathbf{0}$ with $\nabla F(\mathbf{0}) = \mathbf{T} - \boldsymbol{\Delta}$. More generally, $\mathbf{m} = \boldsymbol{\Delta} - \mathbf{L}(q)\mathbf{x}$, linking to $\nabla F(\mathbf{x}, q) = \mathbf{T} - \mathbf{m}$. The condition $m_v > T_v$ is equivalent to $\partial F / \partial x_v < 0$, ensuring updates

---

**Algorithm 2** Query-Aware Flow Diffusion

**Input:** Graph $\mathcal{G} = (\mathcal{V}, \mathcal{E})$ with node embeddings $\{\mathbf{e}_v\}_{v \in \mathcal{V}}$, seed node $s \in \mathcal{V}$ (via Algorithm 1), query $q$ and embedding $\mathbf{e}_q \leftarrow h(q)$, $\alpha, \epsilon \ge 0$.
**Output:** Node importance scores $\mathbf{x} \in \mathbb{R}^{|\mathcal{V}|}$
1: Initialize $\mathbf{m} \leftarrow \mathbf{0}$, $\mathbf{x} \leftarrow \mathbf{0}$
2: Set sink capacities $T_v = \deg(v)$ for all $v \in \mathcal{V}$
3: Set source mass $m_s = \alpha \cdot \sum_{v \in \mathcal{V}} T_v$
4: **while** $\sum_v \max(0, m_v - T_v) > \epsilon$ **do**
5:     Select $v \in \{u : m_u > T_u\}$ u.a.r.
6:     $\text{excess} \leftarrow m_v - T_v, \quad w_v \leftarrow 0$
7:     **for** $u \in \text{neighbors}(v)$ **do**
8:         Compute $\bar{w}(q, v, u)$ using Eq. (1)
9:         $w_v \leftarrow w_v + \bar{w}(q, v, u)$
10:     **end for**
11:     $x_v \leftarrow x_v + \text{excess}/w_v, \quad m_v \leftarrow T_v$
12:     **for** $u \in \text{neighbors}(v)$ **do**
13:         $m_u \leftarrow m_u + \text{excess} \cdot \bar{w}(q, v, u)/w_v$
14:     **end for**
15: **end while**
16: **return** $\mathbf{x}$

decrease the objective. Push operations enforce complementary slackness: active $x_v > 0$ correspond to nodes at capacity ($m_v = T_v$), satisfying dual optimality.

**Locality Preservation and On-Demand Computation.** After selecting seedource nodes $\mathcal{S}_{\mathcal{V}}$, Algorithm 2 preserves the locality of flow diffusion while enabling online graph traversal. This property is crucial for large-scale KGs with millions of nodes. Unlike SOTA RAGs (Edge et al., 2024a), which requires full preprocessing, QAFD-RAG dynamically discovers relevant entities and relationships during traversal. When mass flows from node $v$ to its neighbors, the algorithm: (i) computes query-aware edge weights $\bar{w}(q, v, u)$ on-demand via (1); (ii) retrieves embeddings $\mathbf{e}_u$ and properties only when mass reaches $u$; and (iii) determines sink capacities $T_u$ dynamically. This lazy evaluation makes complexity scale with the explored subgraph rather than the full graph.

**Extension to Multi-Subquery Formulation.** For complex queries requiring multi-hop reasoning, a single embedding via (1) may be insufficient. We address this by decomposing such queries into multiple subqueries, as they often involve distinct reasoning aspects solvable independently and then combined. For example, the Spider 2.0 (Lei et al., 2024) query *"Please help me find the film category with the highest total rental hours in cities whose names either start with 'A' or contain a hyphen."* decomposes into filtering cities, linking to addresses and customers, finding rentals, mapping rentals to films, and identifying film categories with the highest totals. This decomposition enables more effective embeddings during traversal by handling reasoning aspects independently before aggregation. Given a complex query $q$, we decompose it into $K$ subqueries $\mathcal{Q}_K = \{q_1, q_2, \ldots, q_K\}$ using LLM-based decomposition (see Prompt 5), where each $q_i$ captures a specific reasoning aspect. We then apply flow diffusion to each subquery independently: for each $q_k$, we compute

$$\bar{w}(q_k, u, v) = c \cdot H_{\text{sim}}(h(u), h(v)) \circ [a + b \cdot (H_{\text{sim}}(h(u), h(q_k)) \circ H_{\text{sim}}(h(v), h(q_k)))]. \quad (6)$$

We then solve the flow optimization problem for each subquery. From (5), we obtain

$$\min_{\mathbf{x}^{(k)} \in \mathbb{R}_+^{|\mathcal{V}|}} F(\mathbf{x}^{(k)}; q_k) := \tfrac{1}{2}(\mathbf{x}^{(k)})^\top \mathbf{L}^{(k)}(q_k)\mathbf{x}^{(k)} + (\mathbf{x}^{(k)})^\top (\mathbf{T}^{(k)} - \boldsymbol{\Delta}^{(k)}), \qquad \forall q_k \in \mathcal{Q}_K. \quad (7)$$

Here, $\mathbf{L}^{(k)}(q_k) = \mathbf{B}\bar{\mathbf{W}}^{(k)}(q_k)\mathbf{B}^\top$ is the weighted Laplacian for $q_k$, with $\bar{\mathbf{W}}^{(k)}(q_k)$ denoting the diagonal matrix of edge weights $\bar{w}(q_k, u, v)$. The solution yields subquery-specific importance scores $\mathbf{x}^{(k)}$, with support $\mathcal{V}_{\text{support}}^{(k)} = \{v \in \mathcal{V} : x_v^{(k)} > \underline{\epsilon}\}$, where $\underline{\epsilon} > 0$ filters out negligible values. The final retrieved subgraph combines all subqueries, $\mathcal{G}_q = \bigcup_{k=1}^{K} \mathcal{G}_{q_k}$, where each $\mathcal{G}_{q_k}$ is built from $\mathcal{V}_{\text{support}}^{(k)}$ and its edges. See Figure 5 for an illustrative example.

## 3 OPTIMIZATION AND STATISTICAL GUARANTEES

We now provide theoretical guarantees for Algorithm 2, focusing on convergence and locality.

**Lemma 2.** *Let $\mathbf{x}^{(k)*}$ be the optimal solution of (7). The support of each iterate generated by Algorithm 2 is contained within $\text{supp}(\mathbf{x}^{(k)*})$. Moreover, $|\text{supp}(\mathbf{x}^{(k)*})| \leq \|\boldsymbol{\Delta}^{(k)}\|_1$.*

Let $\bar{d}$ be the maximum degree of a node in $\text{supp}(\mathbf{x}^*)$. Since each iteration only touches a node $u \in \text{supp}(\mathbf{x}^*)$ and its neighbors, Lemma 2 implies that the number of nodes ever explored is at most $\|\boldsymbol{\Delta}\|_1$. Thus, if $\|\boldsymbol{\Delta}\|_1$ is small and $\bar{d}$ does not scale linearly with $n$, Algorithm 2 remains local, with subgraph size controlled by $\|\boldsymbol{\Delta}\|_1$.

**Theorem 3.** *For subquery $q_k$, assume $|\text{supp}(\mathbf{x}^{(k)*})| < |\mathcal{V}|$, where $\mathbf{x}^{(k)*}$ solves (7). After $\tau^{(k)} = O\left(\|\boldsymbol{\Delta}^{(k)}\|_1 \frac{\gamma^{(k)}}{\eta^{(k)}} \log \frac{1}{\xi}\right)$, where $\gamma^{(k)} = \max_{u \in \text{supp}(\mathbf{x}^{(k)*})} \sum_{v \sim u} \bar{w}(q_k, u, v)$ and $\eta^{(k)} \leq \min_{(u,v) \in \text{supp}(\mathbf{B}\mathbf{x}^{(k)*})} \bar{w}(q_k, u, v)$, we obtain $\mathbb{E}\left[F\left(\mathbf{x}^{(k),\tau^{(k)}}; q_k\right)\right] - F(\mathbf{x}^{(k)*}; q_k) \leq \xi$.*

**Corollary 4.** *Algorithm 2 converges to $\mathbf{x}^{(k)*}$ in $O(\bar{d} \cdot \|\boldsymbol{\Delta}\|_1 \cdot \log(1/\epsilon))$ iterations, where $\bar{d}$ is the maximum degree in the KG, $\|\boldsymbol{\Delta}\|_1$ is the total injected mass, and $\epsilon$ is the desired accuracy.*

Theorem 3 shows that convergence depends on $\gamma^{(k)}/\eta^{(k)}$, where $\gamma^{(k)}$ is the maximum query-aware weighted degree and $\eta^{(k)}$ the minimum edge weight in the optimal support. Since $\bar{w}(q_k, u, v)$ is query-dependent, convergence adapts to each subquery's semantics—unlike traditional diffusion with static weights. The runtime to reach an $\epsilon$-accurate solution is $O(\bar{d}\|\boldsymbol{\Delta}\|_1 \frac{\alpha}{\beta} \log \frac{1}{\epsilon})$. If $\bar{d}$, $\|\boldsymbol{\Delta}\|_1$, and

$\alpha/\beta$ are sublinear in $n$, Algorithm 2 achieves sublinear time, scaling to large KGs. This improves over exhaustive subgraph and graph-based RAG methods with $O(|\mathcal{V}|^2)$ complexity, showing QAFD-RAG combines efficiency with semantic adaptability.

**Statistical Guarantees under a Random Graph Model.** Under appropriate signal-to-noise conditions, we can provide guarantees about the quality of recovered reasoning paths.

Our analysis relies on a probabilistic framework where the embeddings $\mathbf{e}_u$, $\mathbf{e}_v$, and $\mathbf{e}_{q_k}$ for relevant nodes and queries follow a structured random model. While this assumption may seem restrictive, these embeddings can capture diverse node and query characteristics beyond simple text. The mathematical foundations for analyzing node-attributed graphs have been extensively developed in prior work on graph clustering (Andersen & Peres, 2009; Allen-Zhu et al., 2013; Andersen et al., 2016; Shi et al., 2017; Wang et al., 2017; Fountoulakis et al., 2020a; Liu & Gleich, 2020) and diffusion processes (Fountoulakis et al., 2020b; Yang & Fountoulakis, 2023; Fountoulakis et al., 2021). Our novelty lies in incorporating query semantics into this framework, allowing us to analyze how query-dependent edge weights shape traversal dynamics and to provide the subgraph recovery guarantees.

**Definition 5.** *Given a schema graph with node set $\mathcal{V}$, for each query $q_k$, define $\mathcal{R}_k \subseteq \mathcal{V}$ as the set of relevant nodes (tables/columns required by the query) with $r_k = |\mathcal{R}_k|$. The random graph generation is governed by the following edge probabilities: for each pair of nodes $(u, v)$, edges are independently drawn with probability $\rho_1$ if $u, v \in \mathcal{R}_k$, with probability $\rho_2$ if exactly one node is in $\mathcal{R}_k$, and otherwise follow the original schema structure. For each node $u \in \mathcal{V}$, we have $\mathbf{e}_u = \boldsymbol{\mu}_u + \mathbf{z}_u$, where $\boldsymbol{\mu}_u \in \mathbb{R}^d$ is a deterministic vector, and $\mathbf{z}_u \in \mathbb{R}^d$ is random noise with independent mean-zero sub-Gaussian coordinates $z_{u\ell}$, each having variance proxy $\sigma_\ell$:*

$$\mathrm{Prob}(|z_{u\ell}| \geq t) \leq 2\exp\left(-\frac{t^2}{2\sigma_\ell^2}\right), \quad \text{for all } t \geq 0.$$

*For each $q_k$, let $\mathbf{e}_{q_k} = \boldsymbol{\mu}_{q_k} + \mathbf{z}_{q_k}$, using the same noise model. For nodes in the relevant set $\mathcal{R}_k$, we set $\boldsymbol{\mu}_u = \boldsymbol{\mu}_v = \boldsymbol{\mu}_{q_k}$ for all $u, v \in \mathcal{R}_k$. Finally, for query $q_k$, we define $\bar{w}(q_k, u, v) = \bar{w}_{Product}(q_k, u, v)$, where $H_{sim}(\mathbf{a}, \mathbf{b}) = \exp(-\gamma\|\mathbf{a} - \mathbf{b}\|^2)$ is an RBF-kernel similarity function. We also allow distinct bandwidths $\gamma_i$ for each factor in $\bar{w}_{Product}$.*

Note that this formulation is chosen for simplicity of analysis. In practice, other measures–such as cosine similarity–can be used depending on the application. Throughout, we define

$$\hat{\sigma} := \max_{1 \leq \ell \leq d} \sigma_\ell \quad \text{and} \quad \hat{\mu} := \min_{u \in \mathcal{R}_k, v \notin \mathcal{R}_k} \|\boldsymbol{\mu}_u - \boldsymbol{\mu}_v\|. \tag{8}$$

**Assumption A** (Knowledge Graph Structure and Signal-to-Noise). *The knowledge graph satisfies the following: (1) Entities relevant to a query $q_i$ form a connected subgraph $\mathcal{R}_i$; (2) The embedding space preserves semantic similarity with signal-to-noise ratio $\hat{\mu}/\hat{\sigma} = \omega(\sqrt{d\log|\mathcal{V}|})$; (3) Variance concentrates as $\sum_{\ell=1}^d \sigma_\ell^2/\hat{\sigma}^2 = O(\log|\mathcal{V}|)$; and (4) Irrelevant entities have embeddings well-separated from query embeddings.*

**Lemma 6** (Query-Aware Edge Weight Separation). *Under Assumption A, if the similarity function parameters satisfy $\gamma_i\hat{\sigma}^2 = o(\log^{-1}|\mathcal{V}|)$ for $i \in \{1, 2, 3\}$, then with probability at least $1 - O(|\mathcal{V}|^{-2})$: (i) For all $u, v \in \mathcal{R}_k$, $\bar{w}(q_k, u, v) \geq (1 - o(1))$; (ii) For all $u \in \mathcal{R}_k, v \in \mathcal{V} \setminus \mathcal{R}_k$, $\bar{w}(q_k, u, v) \leq \exp(-\omega(\log|\mathcal{V}|))$.*

Lemma 6 shows that under the signal-to-noise conditions in Assumption A, query-aware edge weights $\bar{w}(q_k, u, v)$ separate relevant nodes $\mathcal{R}_k$ from irrelevant ones $\mathcal{V} \setminus \mathcal{R}_k$. With similarity parameters $\gamma_i$ scaled inversely with $\log|\mathcal{V}|$, the lemma guarantees with high probability that (i) edges within $\mathcal{R}_k$ remain essentially unchanged, preserving reasoning paths, while (ii) edges between relevant and irrelevant nodes become exponentially small, blocking diffusion into contextually irrelevant regions.

**Theorem 7.** *Under the conditions of Lemma 6, assume that either (i) the induced subgraph on $\mathcal{R}_k$ is connected, or (ii) $\rho_1 \geq \frac{(4+\epsilon)\log r_k}{\delta^2(r_k-1)}$ for some $\delta \in (0, 1)$ and $\epsilon > 0$. If the source mass is set as $\Delta_s^{(k)} = (1 + \beta)\sum_{u \in \mathcal{R}_k} T_u$ for any $\beta > 0$, then with high probability we have*

$$\mathcal{R}_k \subseteq \mathrm{supp}(\mathbf{x}^{(k)}), \quad \text{and} \quad \sum_{u \in \mathrm{supp}(\mathbf{x}^{(k)})\setminus\mathcal{R}_k} T_u \leq \beta\sum_{u \in \mathcal{R}_k} T_u. \tag{9}$$

Table 1: Comparison of QAFD-RAG and baselines on UltraDomain across five GPT-4o–scored dimensions (0–100). Rows are grouped by dataset and columns by metric; values are mean ($\pm$ std) over 5 evaluations. Best per dataset/metric is bolded. *Continued in Appendix A2.1.1.*

| Dataset | Method | Comprehensive. | Diversity | Logicality | Relevance | Coherence |
|---|---|---|---|---|---|---|
| Agriculture | GraphRAG | 87.30 ($\pm$4.46) | 82.85 ($\pm$4.73) | 90.80 ($\pm$5.84) | 94.01 ($\pm$6.62) | 90.08 ($\pm$3.23) |
| | LightRAG | 83.65 ($\pm$5.97) | 77.71 ($\pm$7.34) | 88.85 ($\pm$3.76) | 93.55 ($\pm$4.16) | 88.67 ($\pm$2.57) |
| | RAPTOR | 83.32 ($\pm$8.67) | 76.65 ($\pm$12.08) | 89.54 ($\pm$3.63) | 94.56 ($\pm$3.41) | 89.47 ($\pm$3.01) |
| | HippoRAG | 82.51 ($\pm$5.14) | 76.26 ($\pm$9.23) | 88.84 ($\pm$3.26) | 94.09 ($\pm$2.48) | 88.79 ($\pm$2.73) |
| | QAFD-RAG | **89.93 ($\pm$3.36)** | **84.95 ($\pm$4.26)** | **92.10 ($\pm$2.53)** | **95.67 ($\pm$3.28)** | **92.00 ($\pm$1.62)** |
| Biology | GraphRAG | 85.76 ($\pm$10.80) | 81.05 ($\pm$10.39) | 88.94 ($\pm$11.70) | 93.00 ($\pm$12.50) | 88.57 ($\pm$11.10) |
| | LightRAG | 83.92 ($\pm$4.13) | 78.28 ($\pm$6.46) | 88.40 ($\pm$4.24) | 93.31 ($\pm$5.42) | 88.09 ($\pm$2.86) |
| | RAPTOR | 83.57 ($\pm$6.20) | 77.10 ($\pm$9.41) | 88.33 ($\pm$5.62) | 93.62 ($\pm$6.42) | 88.67 ($\pm$4.21) |
| | HippoRAG | 83.07 ($\pm$4.15) | 76.91 ($\pm$7.25) | 88.07 ($\pm$3.69) | 93.62 ($\pm$3.53) | 88.52 ($\pm$2.90) |
| | QAFD-RAG | **89.44 ($\pm$3.92)** | **85.13 ($\pm$4.11)** | **91.19 ($\pm$4.20)** | **95.05 ($\pm$4.71)** | **91.33 ($\pm$2.59)** |
| Cooking | GraphRAG | 86.23 ($\pm$7.00) | 79.10 ($\pm$8.72) | 90.79 ($\pm$2.68) | 95.14 ($\pm$2.91) | 90.63 ($\pm$1.75) |
| | LightRAG | 82.11 ($\pm$7.56) | 74.97 ($\pm$9.46) | 87.49 ($\pm$5.82) | 92.27 ($\pm$5.88) | 87.98 ($\pm$4.09) |
| | RAPTOR | 83.15 ($\pm$6.99) | 75.30 ($\pm$9.79) | 88.52 ($\pm$5.12) | 93.59 ($\pm$5.48) | 88.83 ($\pm$3.87) |
| | HippoRAG | 82.52 ($\pm$4.55) | 74.13 ($\pm$7.25) | 88.40 ($\pm$3.20) | 93.71 ($\pm$2.66) | 88.57 ($\pm$2.61) |
| | QAFD-RAG | **89.25 ($\pm$3.82)** | **83.42 ($\pm$5.25)** | **91.35 ($\pm$2.73)** | **95.45 ($\pm$2.83)** | **91.58 ($\pm$2.04)** |
| History | GraphRAG | 84.18 ($\pm$10.30) | 78.88 ($\pm$9.97) | 88.18 ($\pm$11.63) | 92.35 ($\pm$13.51) | 88.40 ($\pm$10.71) |
| | LightRAG | 82.24 ($\pm$6.04) | 76.42 ($\pm$7.51) | 86.98 ($\pm$5.75) | 92.18 ($\pm$6.17) | 87.18 ($\pm$4.24) |
| | RAPTOR | 82.08 ($\pm$7.43) | 75.59 ($\pm$9.75) | 87.67 ($\pm$4.63) | 92.94 ($\pm$4.93) | 88.13 ($\pm$3.58) |
| | HippoRAG | 80.45 ($\pm$6.95) | 74.61 ($\pm$8.22) | 86.37 ($\pm$6.36) | 91.54 ($\pm$8.22) | 86.97 ($\pm$4.77) |
| | QAFD-RAG | **87.75 ($\pm$3.96)** | **83.14 ($\pm$4.40)** | **90.04 ($\pm$3.93)** | **93.77 ($\pm$6.25)** | **90.55 ($\pm$2.37)** |
| Legal | GraphRAG | 84.96 ($\pm$9.72) | **78.74 ($\pm$10.28)** | 88.67 ($\pm$8.58) | 91.01 ($\pm$10.90) | 88.44 ($\pm$5.77) |
| | LightRAG | 79.63 ($\pm$11.25) | 67.63 ($\pm$11.86) | 86.06 ($\pm$7.47) | 90.77 ($\pm$10.17) | 86.12 ($\pm$6.10) |
| | RAPTOR | 81.43 ($\pm$9.67) | 64.97 ($\pm$13.91) | 88.34 ($\pm$6.54) | 93.29 ($\pm$9.18) | 87.70 ($\pm$6.16) |
| | HippoRAG | 82.23 ($\pm$8.85) | 64.28 ($\pm$11.31) | 88.56 ($\pm$7.64) | **93.60 ($\pm$9.45)** | 87.95 ($\pm$6.39) |
| | QAFD-RAG | **86.19 ($\pm$5.86)** | 77.14 ($\pm$7.42) | **90.06 ($\pm$5.06)** | 93.30 ($\pm$9.66) | **89.99 ($\pm$3.35)** |

Theorem 7 shows that query-aware flow diffusion reliably recovers the query-relevant subgraph $\mathcal{R}_k$ with controlled precision. It provides two guarantees: **Complete recovery** ($\mathcal{R}_k \subseteq \text{supp}(\mathbf{x}^{(k)})$) ensures all semantically relevant nodes to query $q_k$ receive positive flow and are included, so no essential information is lost. **Limited leakage** ($\sum_{u \in \text{supp}(\mathbf{x}^{(k)}) \setminus \mathcal{R}_k} T_u \leq \beta \sum_{u \in \mathcal{R}_k} T_u$) bounds flow escaping into irrelevant regions. The trade-off parameter $\beta$ controls this balance: smaller $\beta$ enforces tighter focus, while larger $\beta$ allows peripheral context at some cost to precision. Together, these results show that QAFD-RAG's query-aware edge weighting acts as a semantic filter, concentrating flow on relevant reasoning paths while minimizing diffusion to unrelated regions.

## 4 EXPERIMENTAL EVALUATION

We evaluate QAFD-RAG against strong baselines on knowledge graph question-answering and text-to-SQL benchmarks, with a particular emphasis on query-aware retrieval performance and schema linking capabilities.

### 4.1 KNOWLEDGE GRAPH QUESTION ANSWERING RESULTS

#### 4.1.1 BASELINES FOR KG-QA

We compare **QAFD-RAG** with four recent graph-oriented RAG systems under an identical setup (corpora, prompts, evaluation): **GraphRAG** (Edge et al., 2024b), which clusters retrieved documents via community detection and builds hierarchical contexts; **LightRAG** (Guo et al., 2024), which performs dual-level retrieval in the indexing graph; **HippoRAG** (Jimenez Gutierrez et al., 2024), which constructs a passage-level KG and applies Personalized PageRank for single-step multi-hop retrieval; and **RAPTOR** (Sarthi et al., 2024), which forms a hierarchical retrieval tree through recursive clustering and abstractive summarization, enabling multi-level evidence selection.

Table 2: Performance comparison of QAFD-RAG against baseline methods on SQuALITY long-document summarization.

| Method | BLEU-1 | BLEU-2 | ROUGE-1 F1 | ROUGE-2 F1 | METEOR |
|--------|--------|--------|------------|------------|--------|
| GraphRAG | 33.91 | 16.12 | 26.38 | 4.08 | 24.38 |
| HippoRAG | 33.22 | 16.74 | 27.29 | 3.92 | 23.41 |
| RAPTOR | 32.10 | 16.58 | 25.13 | 3.49 | 22.87 |
| LightRAG | 34.17 | 17.41 | **28.59** | 4.31 | 23.27 |
| QAFD-RAG | **35.44** | **18.63** | 28.43 | **4.79** | **25.59** |

### 4.1.2 LLM CONFIGURATION AND HYPERPARAMETERS

Unless otherwise noted, we use `GPT-4o-mini` for all entity/relation extraction, hierarchical key-wording, and cluster summarization, `text-embedding-3-small` (1536d) for all embeddings, and a fixed configuration of 40 seed nodes, diffusion mass $\alpha = 50$, and hybrid query-aware edge weights (Eq. (2c)) with $a = 1$ and $b = \frac{1}{4}$. Additional implementation details are provided in Appendix A4.

### 4.1.3 DATASETS AND EVALUATION PROTOCOL

We evaluate QAFD-RAG across three major settings. **General QA** uses ten UltraDomain subsets (Qian et al., 2024) spanning long-context reasoning and implicit inference (Agriculture, Biology, Cooking, History, Legal, Mathematics, Mix, Music, Philosophy, Physics). Answers are scored along five axes—*Comprehensiveness, Diversity, Logicality, Relevance, Coherence*—using GPT-4o on a $0-100$ scale, with five independent evaluations per response (Appendix A4). For **Long-Document Summarization**, we use SQuALITY (Wang et al., 2022), where answers are question-focused abstractive summaries benchmarked against multiple human-written references; we report *BLEU-1*, *BLEU-2*, *ROUGE-1-F1*, *ROUGE-2-F1*, and *METEOR* (Appendix A2.1.5). **Multi-Hop QA** is evaluated on HotpotQA (Yang et al., 2018), 2WikiMultiHopQA (Ho et al., 2020), and MuSiQue (Trivedi et al., 2022) which require combining evidence across documents and explicit chains; we report *Exact Match (EM)* and *F1* using official evaluation scripts (Appendix A4.1).

### 4.1.4 EXPECTED RESULTS AND ANALYSIS

On the ten ULTRADOMAIN subsets (Tables 1, 5), QAFD-RAG achieves the strongest overall averages across five axes, with steady gains in *Comprehensiveness*, *Logicality*, and *Coherence*. Baselines show complementary strengths—RAPTOR may excel on *Relevance*/*Coherence* when hierarchical abstraction fits the domain, HippoRAG performs well on *Relevance* with local evidence, and GraphRAG/LightRAG can boost *Diversity* by expanding neighborhoods, often at the cost of *Logicality*. Overall, focusing retrieval on a query-aligned subgraph improves factual alignment without hurting fluency, whereas baselines trade off precision and recall in domain-dependent ways.

On the SQuALITY benchmark (Table 2), QAFD-RAG achieves the highest performance on the majority of automated metrics—including BLEU-1, BLEU-2, ROUGE-2 F1, and METEOR—indicating more faithful and coherent long-form summaries compared to GraphRAG, HippoRAG, RAPTOR, and LightRAG. While LightRAG now attains the highest ROUGE-1 F1 score, QAFD-RAG's query-aligned retrieval yields a stronger overall metric profile, offering a more stable balance between precision and coverage across diverse narrative questions. On HotpotQA, MuSiQue, and 2WikiMultiHopQA, QAFD-RAG achieves the highest *F1* and *EM* scores on HotpotQA and MuSiQue, and remains competitive with HippoRAG on 2WikiMultiHopQA (Table 3). These improvements in *F1* and *EM* highlight that query-aware diffusion more reliably recovers gold evidence under strict matching. While HippoRAG also demonstrates strong multi-hop reasoning through personalized PageRank, QAFD-RAG's query-aligned pruning produces tighter evidence chains and a stable precision–recall balance, enabling superior or competitive performance across multi-hop reasoning tasks.

## 4.2 TEXT-TO-SQL RESULTS

We compare against methods from different paradigms. **CHASE-SQL** (Pourreza et al., 2024) uses multi-path LLM reasoning (divide-and-conquer, execution-plan CoT, instance-aware few-shot) with

Table 3: Performance on Multi-Hop QA (F1, EM).

| Method | HotpotQA | | 2WikiMultihopQA | | MuSiQue | |
|---|---|---|---|---|---|---|
| | F1 | EM | F1 | EM | F1 | EM |
| GraphRAG | 33.40 | 14.00 | 15.20 | 7.00 | 39.40 | 17.60 |
| LightRAG | 8.80 | 0.00 | 8.20 | 1.00 | 1.40 | 0.10 |
| RAPTOR | 52.30 | 25.00 | 38.80 | 12.00 | 28.10 | 10.50 |
| HippoRAG | 58.67 | 45.27 | **70.33** | **61.16** | 38.33 | 27.55 |
| QAFD-RAG | **73.42** | **58.10** | 69.41 | 59.50 | **47.99** | **33.50** |

a pairwise-selection agent for SQL generation and ranking. **DIN-SQL** (Pourreza & Rafiei, 2024) adopts a decomposition framework with template matching. **Spider-Agent** (Lei et al., 2024) applies ReAct-style schema exploration and iterative SQL generation. **CodeS** (Li et al., 2024b) emphasizes code generation via prompt engineering. **DAIL-SQL** (Gao et al., 2023a) improves Text-to-SQL accuracy by selecting and arranging few-shot code examples using masked question–query similarity. All methods are using GPT-4o LLM calls.

We evaluate on the Spider 2.0 (Lei et al., 2024), a comprehensive dataset with diverse schema architectures representative of real enterprise environments. The benchmark includes complex SQL queries requiring multi-table reasoning and advanced analytical operations. Our analysis focuses on the *Spider 2.0 Local Test Set*, which contains 135 SQLite and Snowflake instances featuring long join chains and intricate query structures. Table 4 reports execution accuracy on both test sets. Here, SQL-Agent is a variant of the Spider-Agent (Lei et al., 2024)

Table 4: SQL execution accuracy on 135 SQLite and Snowflake test sets.

| Method | SQLite (%) | Snowflake (%) |
|---|---|---|
| CHASE-SQL | 11.85 | 5.92 |
| DIN-SQL | 2.74 | 0.18 |
| DAIL-SQL | 0.00 | 0.00 |
| CodeS | 0.00 | 0.00 |
| Spider-Agent | 21.50 | 16.30 |
| **QAFD-RAG + SQL-Agent** | **26.70** | **23.70** |

SQL generator. Unlike Spider-Agent, which relies on a ReAct-style agent for exploration and reasoning, our method converts each database into a knowledge graph and applies QAFD to identify relevant tables and columns. SQL-Agent then uses tailored prompts and QAFD-RAG's subgraphs for SQL generation. An example prompt is shown in Prompt 6.

## 5 CONCLUSION, LIMITATIONS, AND FUTURE WORK

We presented QAFD-RAG, a training-free framework with query-aware edge reweighting for adaptive graph traversal. Our contributions include: (1) the first principled flow diffusion for graph-based RAG, (2) theoretical guarantees for subgraph recovery, and (3) consistent improvements across benchmarks while reducing LLM calls. QAFD-RAG relies on pre-trained embeddings, which may underperform in domain-specific settings. Besides, the embedding-based flow diffusion may struggle with explicit logical negation because embeddings capture semantic relatedness rather than logical operators. While QAFD-RAG partially mitigates this through LLM-based keyword extraction (Prompt 2) and response filtering (Prompt 3), this remains an important challenge.

QAFD-RAG's modularity enables drop-in replacement of retrieval components in existing systems: it can replace static Leiden clustering in GraphRAG, enhance Personalized PageRank in HippoRAG with query-aware edge weighting, and extend LightRAG's single-hop retrieval to multi-hop flow diffusion—all without retraining while providing theoretical guarantees. Future work includes learning edge weights from query–answer pairs and extending to temporal/multi-modal graphs. Future work should also explore contrastive embeddings or hybrid symbolic–neural approaches that encode logical distinctions during graph traversal.

ACKNOWLEDGEMENTS

We would like to thank Samsung SDS Research America for supporting this work. We are also grateful to the area chair and the anonymous reviewers for their constructive feedback and valuable suggestions, which helped improve the quality of this paper.

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

# Appendix

CONTENTS

## A1 EXTENDED RELATED WORK

### A1.1 RETRIEVAL-AUGMENTED GENERATION.

LLMs remain prone to hallucinations and a lack of domain knowledge (Sahoo et al., 2024; Wang et al., 2025). Text-based RAG reduces these issues by supplementing LLMs with unstructured external data (Lewis et al., 2020; Guu et al., 2020). These systems employ sparse or dense retrieval (Alon et al., 2022; Schick et al., 2023; Jiang et al., 2023; Cheng et al., 2024; Hofstätter et al., 2023; Li et al., 2024a; Zhang et al., 2024a), but most treat text as flat segments, missing critical context and inter-document relationships (Edge et al., 2024a; Guo et al., 2024).

KG-RAG enhances interpretability and factuality by leveraging structured knowledge graphs (Yasunaga et al., 2021a; Gao et al., 2022; Li et al., 2024c; He et al., 2025). These systems utilize curated (Wen et al., 2023; Dehghan et al., 2024) or optimized (Fang et al., 2024b; Panda et al., 2024) graphs to retrieve entity and relational context, typically extracting local subgraphs relevant to a query (Bordes et al., 2015; Talmor & Berant, 2018; Gu et al., 2021). However, most KG-RAG methods focus on single-hop or shallow queries (Joshi et al., 2017; Yang et al., 2018; Kwiatkowski et al., 2019; Ho et al., 2020) and struggle with scale and multi-step reasoning.

Among training-intensive approaches, QA-GNN (Yasunaga et al., 2021b) combines pre-trained language models and knowledge graphs by using LM-based relevance scoring to select pertinent KG nodes, followed by joint graph neural reasoning for accurate and interpretable question answering. Xiong et al. (2019) proposed knowledge-aware neural retrievers for incomplete KBs. Other works such as SubgraphRAG (Li et al., 2024c) train end-to-end retrieval modules to extract relevant KG subgraphs for downstream LLM reasoning. Several studies have also integrated knowledge graph structure directly into transformers for enhanced QA (Hu et al., 2022), and KnowGPT (Zhang et al., 2024b) leverages KG-based prompting for large language models. GNN-RAG (Mavromatis & Karypis, 2024) trains GNNs to score answer candidates and retrieve shortest paths, while RoG (LUO et al., 2024) uses LLM prompting to generate relation paths. Both methods incorporate query semantics but require training/finetuning, lack theoretical guarantees, and operate over static graph structures during retrieval. However, these methods require substantial supervised data and retraining, in contrast to our training-free, query-aware flow diffusion framework with statistical guarantees.

Recent work has explored training-free KG-RAG methods, building text-associated graphs to support more complex and multi-hop queries (Edge et al., 2024a; Guo et al., 2024). For instance, GraphRAG (Edge et al., 2024a) applies community detection to cluster entities, while LightRAG (Guo et al., 2024) uses multi-stage retrieval and ego-network aggregation. PathRAG (Chen et al., 2025a) further improves graph-based RAG by retrieving key relational paths rather than redundant neighborhood information, using flow-based pruning to identify reliable paths and strategic prompt organization to enhance LLM responses. However, all aforementioned methods still struggle to precisely align query intent with relevant regions of the graph, making it difficult to identify semantically coherent reasoning subgraphs. Furthermore, existing RAG approaches generally lack statistical or optimization guarantees, as well as complexity analysis, for their retrieval mechanisms.

### A1.2 GRAPH DIFFUSION AND FLOW METHODS

Graph diffusion describes the process of spreading mass from one or more seed nodes to neighboring nodes along graph edges. The empirical and theoretical performance of local diffusion methods is typically evaluated in the context of local graph clustering. Local graph clustering was introduced by Spielman & Teng (2013a) using a random-walk algorithm, with Andersen et al. (2006) later employing personalized PageRank. Most works analyze these methods via output conductance (Andersen et al., 2006; Andersen & Peres, 2009; Spielman & Teng, 2013a; Allen-Zhu et al., 2013; Andersen et al., 2016; Shi et al., 2017; Wang et al., 2017; Fountoulakis et al., 2020a; Liu & Gleich, 2020). Statistical analysis appeared in Ha et al. (2021) for $\ell_1$-regularized PageRank (Fountoulakis et al., 2017), though attributed graphs remain unaddressed.

Community/cluster detection methods that combine structure and node or edge attributes (Yang et al., 2013; Jia et al., 2017; Zhe et al., 2019; Sun et al., 2020) benefit from this integration but require global processing, making them unsuitable for local clustering. Contextual random graph models have been employed for attributed community detection (Deshpande et al., 2018; Yan & Sarkar, 2021; Braun et al., 2022; Abbe et al., 2022), node separability (Baranwal et al., 2021; Fountoulakis et al.,

2023a; Baranwal et al., 2023a), and analysis of graph convolutions and optimal classifiers (Wu et al., 2023; Wei et al., 2022; Baranwal et al., 2023b). Related anomaly detection (Arias-Castro et al., 2008; 2011; Sharpnack et al., 2013; Qian & Saligrama, 2014) and estimation (Chitra et al., 2021) methods focus on scalar data and global processing, contrasting with our local, attribute-aware approach.

### A1.3   QAFD-RAG'S POSITIONING WITHIN GRAPH-BASED RAG APPROACHES

To our knowledge, none of these works connect the graph structure to the query. Our work is the first to provide a principled framework that links a given query to the corresponding subgraph in a knowledge graph, and the first to develop and statistically analyze query-aware flow diffusion in general contextual random graph models with formal recovery guarantees.

Beyond these theoretical contributions, QAFD-RAG's modular design positions it as a complementary retrieval component for existing graph-based RAG systems, requiring no retraining while providing formal guarantees. Specifically:

**GraphRAG.** GraphRAG (Edge et al., 2024a) relies on static community detection (Leiden algorithm) to precompute hierarchical clusters and generate community summaries for retrieval. QAFD-RAG offers an alternative through dynamic, query-time subgraph discovery tailored to each query's semantics, eliminating the computational overhead of community summary generation while adapting retrieval to query-specific needs.

**HippoRAG.** HippoRAG (Jimenez Gutierrez et al., 2024) uses Personalized PageRank (PPR) for graph traversal from seed nodes. QAFD-RAG provides a principled alternative through query-aware edge reweighting (Equations (4)-(5)). While PPR provides graph-based signals, it lacks query-aware edge modulation—our method's key innovation that suppresses irrelevant paths while amplifying semantically aligned connections, backed by formal recovery guarantees (Theorem 7).

**LightRAG.** LightRAG (Guo et al., 2024) employs dual-level keyword extraction with single-hop neighborhood aggregation. QAFD-RAG extends this paradigm by enabling multi-hop flow diffusion from extracted keywords as seed nodes, discovering reasoning paths that span multiple hops—precisely the capability that LightRAG's current single-hop approach lacks—while preserving its efficient indexing structure.

This positioning demonstrates QAFD-RAG's role as a foundational component that addresses key limitations in existing graph-based RAG approaches through principled, theoretically-grounded retrieval.

## A2   EXTENDED EXPERIMENTS

### A2.1   QUESTION ANSWERING (QA)

#### A2.1.1   ADDITIONAL NUMERICAL RESULTS FOR GENERAL QUESTION ANSWERING

Table 5 presents comprehensive results across five additional UltraDomain datasets—Mathematics, Mix, Music, Philosophy, and Physics—evaluating QAFD-RAG against GraphRAG, LightRAG, RAPTOR, and HippoRAG across all five GPT-4o-scored dimensions (Comprehensiveness, Diversity, Logicality, Relevance, and Coherence).

QAFD-RAG achieves the best performance across all dimensions in four out of five datasets (Mathematics, Mix, Philosophy, and Physics), demonstrating consistent superiority with scores typically 3–7 points higher than the strongest baseline. For example, on Physics, QAFD-RAG achieves 89.51 Comprehensiveness compared to GraphRAG's 86.33, and 95.61 Relevance compared to RAPTOR's 94.46. On Music, QAFD-RAG leads in four dimensions (Comprehensiveness: 87.95, Diversity: 83.40, Logicality: 90.56, Coherence: 90.94), with GraphRAG marginally ahead only on Relevance (94.14 vs. 94.08). These results validate QAFD-RAG's robustness across diverse domains spanning technical (Mathematics, Physics), creative (Music), conceptual (Philosophy), and mixed content, consistently outperforming state-of-the-art graph-based RAG methods through query-aware flow diffusion and dynamic edge reweighting.

Table 5: Comparison of QAFD-RAG and baselines across five GPT-4o–scored dimensions (0–100). Rows are grouped by dataset; columns are metrics. Each cell shows mean ($\pm$ std) over 5 independent evaluations. Best scores per dataset/metric are bolded. *(continued)*.

| Dataset | Method | Comprehensive. | Diversity | Logicality | Relevance | Coherence |
|---|---|---|---|---|---|---|
| Mathematics | GraphRAG | 84.30 ($\pm$9.52) | 77.46 ($\pm$10.68) | 88.91 ($\pm$8.29) | 92.65 ($\pm$10.34) | 89.04 ($\pm$7.52) |
| | LightRAG | 80.93 ($\pm$8.44) | 74.07 ($\pm$10.40) | 86.30 ($\pm$6.20) | 90.77 ($\pm$8.26) | 86.29 ($\pm$4.56) |
| | RAPTOR | 81.18 ($\pm$9.75) | 73.73 ($\pm$12.34) | 87.30 ($\pm$6.46) | 91.88 ($\pm$8.73) | 87.70 ($\pm$4.57) |
| | HippoRAG | 80.77 ($\pm$5.31) | 73.05 ($\pm$9.06) | 87.05 ($\pm$4.66) | 92.34 ($\pm$5.53) | 87.23 ($\pm$3.68) |
| | QAFD-RAG | **87.30 ($\pm$4.94)** | **82.56 ($\pm$6.09)** | **90.04 ($\pm$5.15)** | **93.36 ($\pm$7.24)** | **90.37 ($\pm$3.16)** |
| Mix | GraphRAG | 82.76 ($\pm$11.41) | 74.94 ($\pm$11.68) | 88.45 ($\pm$6.95) | 92.81 ($\pm$8.65) | 88.90 ($\pm$3.34) |
| | LightRAG | 78.72 ($\pm$13.12) | 70.97 ($\pm$13.39) | 85.72 ($\pm$7.00) | 90.37 ($\pm$9.36) | 86.26 ($\pm$4.68) |
| | RAPTOR | 81.97 ($\pm$5.50) | 73.09 ($\pm$7.75) | 87.90 ($\pm$3.71) | 93.16 ($\pm$3.47) | 87.81 ($\pm$3.28) |
| | HippoRAG | 80.15 ($\pm$5.20) | 71.71 ($\pm$6.12) | 86.55 ($\pm$3.97) | 92.11 ($\pm$3.56) | 86.83 ($\pm$3.35) |
| | QAFD-RAG | **87.15 ($\pm$3.46)** | **81.15 ($\pm$4.86)** | **90.70 ($\pm$2.93)** | **94.36 ($\pm$4.50)** | **90.36 ($\pm$2.07)** |
| Music | GraphRAG | 85.32 ($\pm$6.46) | 79.37 ($\pm$7.92) | 90.02 ($\pm$4.63) | **94.14 ($\pm$7.01)** | 89.80 ($\pm$3.25) |
| | LightRAG | 81.14 ($\pm$8.54) | 75.08 ($\pm$10.13) | 86.64 ($\pm$6.29) | 91.29 ($\pm$8.41) | 87.34 ($\pm$4.36) |
| | RAPTOR | 81.44 ($\pm$8.20) | 74.55 ($\pm$11.26) | 87.81 ($\pm$4.36) | 93.24 ($\pm$3.91) | 88.14 ($\pm$3.54) |
| | HippoRAG | 80.95 ($\pm$6.00) | 74.46 ($\pm$8.33) | 87.22 ($\pm$3.98) | 92.47 ($\pm$4.76) | 87.53 ($\pm$3.67) |
| | QAFD-RAG | **87.95 ($\pm$3.99)** | **83.40 ($\pm$4.47)** | **90.56 ($\pm$3.77)** | 94.08 ($\pm$5.46) | **90.94 ($\pm$2.39)** |
| Philosophy | GraphRAG | 84.61 ($\pm$8.41) | 78.36 ($\pm$8.90) | 88.67 ($\pm$6.90) | 93.53 ($\pm$8.05) | 88.83 ($\pm$6.59) |
| | LightRAG | 80.92 ($\pm$8.88) | 74.36 ($\pm$9.63) | 85.74 ($\pm$6.57) | 90.85 ($\pm$7.32) | 86.44 ($\pm$4.61) |
| | RAPTOR | 82.30 ($\pm$5.97) | 75.54 ($\pm$7.46) | 87.30 ($\pm$4.60) | 92.83 ($\pm$4.65) | 87.83 ($\pm$3.45) |
| | HippoRAG | 80.93 ($\pm$5.46) | 74.12 ($\pm$6.75) | 86.66 ($\pm$4.52) | 92.03 ($\pm$4.69) | 87.42 ($\pm$3.38) |
| | QAFD-RAG | **86.78 ($\pm$4.11)** | **81.91 ($\pm$4.69)** | **89.35 ($\pm$4.25)** | **93.63 ($\pm$5.91)** | **89.91 ($\pm$2.70)** |
| Physics | GraphRAG | 86.33 ($\pm$7.32) | 79.10 ($\pm$7.84) | 90.29 ($\pm$7.68) | 94.73 ($\pm$7.71) | 89.99 ($\pm$7.37) |
| | LightRAG | 84.45 ($\pm$4.39) | 76.38 ($\pm$6.19) | 89.13 ($\pm$4.37) | 93.67 ($\pm$5.39) | 88.65 ($\pm$2.76) |
| | RAPTOR | 84.18 ($\pm$3.98) | 75.31 ($\pm$6.27) | 89.32 ($\pm$3.52) | 94.46 ($\pm$2.81) | 89.22 ($\pm$2.92) |
| | HippoRAG | 82.81 ($\pm$3.63) | 73.66 ($\pm$5.81) | 88.66 ($\pm$3.07) | 94.09 ($\pm$2.43) | 88.69 ($\pm$2.54) |
| | QAFD-RAG | **89.51 ($\pm$3.14)** | **84.21 ($\pm$3.89)** | **91.77 ($\pm$3.26)** | **95.61 ($\pm$2.74)** | **91.67 ($\pm$2.14)** |

## A2.1.2   SENSITIVITY ANALYSIS FOR QAFD-RAG

To assess the robustness of QAFD-RAG under different configurations, we conduct a sensitivity analysis on three key hyperparameters on the Mix dataset in our framework: the number of seed nodes used for QAFD, the initial mass parameter $\alpha$ in the QAFD process, and the choice of edge weight formulation that governs propagation dynamics. We define the tuning ranges as follows:

- Number of seed nodes: $\{20, 30, 40, 50, 60\}$
- $\alpha$ (mass initialization): $\{5, 10, 20, 50, 100\}$
- Query-aware edge weight: three distinct formulations reflecting different query-to-node affinity strategies (as presented in Eqn (2a), (2b), and (2c)).

When analyzing a single hyperparameter, the others are held constant at their default values: $\alpha = 50$, the number of seed nodes is 20, and the default query-aware edge weight function is the hybrid formulation.

Figure 3 shows that overall performance is stable across the tested ranges, confirming the robustness of QAFD-RAG. Among all metrics, only relevance exhibits a noticeable dependency on the number of seed nodes, slightly improving as fewer seed nodes are introduced, suggesting that 20 seed nodes are already sufficient for most evaluation dimensions. In contrast, $\alpha = 50$ yields the best overall balance across metrics; both very small and very large values tend to slightly degrade performance, likely due to insufficient or overly diffused mass concentration during propagation. Surprisingly, the choice of edge weight function has a relatively minor effect—all three variants yield comparable results across metrics, with only slight differences in relevance and logicality.

These findings indicate that QAFD-RAG is not overly sensitive to hyperparameter settings and performs robustly across a wide range of configurations, requiring minimal tuning in practice.

## A2.1.3   RUNTIME AND EFFICIENCY OF QAFD-RAG FOR GENERAL QA

Table 6 shows that QAFD-RAG matches or outperforms LightRAG in wall-clock time and is substantially faster than GraphRAG. The efficiency gains primarily come from indexing: instead of

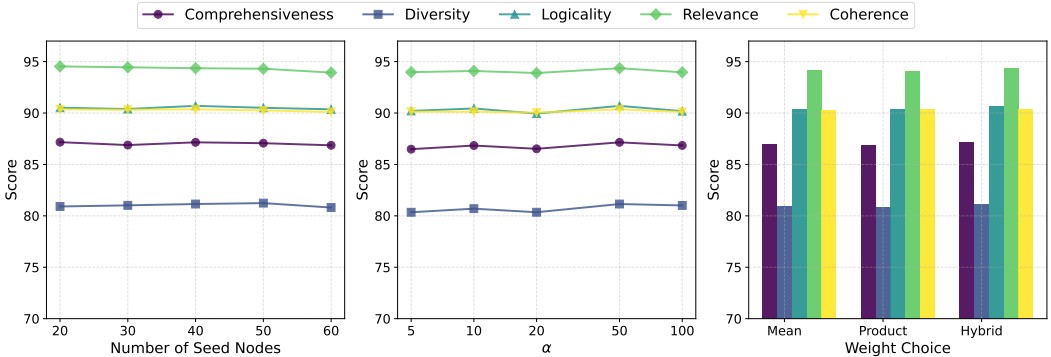

Figure 3: Sensitivity of QAFD-RAG to the number of seed nodes (left), initial mass $\alpha$ (middle), and edge weight choice (right), evaluated across five dimensions on the Mix dataset.

Table 6: Total runtime (in seconds) for graph-based RAG on the full Mix dataset.

| Method | QAFD-RAG | LightRAG | GraphRAG |
|---|---|---|---|
| **Total Time (s)** | 8948.81 | 9236.38 | 50968.59 |

performing global or hierarchical summarization, QAFD-RAG processes only the subgraph retrieved at query time. The diffusion step adds negligible overhead because edge weights are computed on demand and traversal is nearly linear. Overall, these properties provide a favorable balance of accuracy and efficiency.

### A2.1.4  EMBEDDING MODEL SENSITIVITY ANALYSIS

To assess QAFD-RAG's robustness to embedding quality, we evaluate the framework across five diverse embedding models on the Mix dataset from UltraDomain. We compare: (i) OpenAI's `text-embedding-3-small` (1536-dim) and `text-embedding-3-large` (3072-dim), representing cloud-based proprietary embeddings; (ii) Jina AI's `jina-embeddings-v3` (Sturua et al., 2024) (1024-dim), an open-source local model; (iii) NVIDIA's `nv-embed-v2` (Lee et al., 2024) (4096-dim), optimized for retrieval tasks; and (iv) `GritLM-7B` (Muennighoff et al., 2024) (4096-dim), a unified generative-embedding model. All other hyperparameters remain fixed across experiments.

Table 7 presents results across five evaluation dimensions. QAFD-RAG demonstrates consistent performance across all embedding models, with Comprehensiveness scores ranging from 85.82 to 88.12 and Relevance scores from 91.98 to 95.12. Notably, `text-embedding-3-large` achieves the best performance on Comprehensiveness, Diversity, and Relevance, while `nv-embed-v2` excels on Logicality and Coherence. However, the overlapping standard deviations across models indicate that differences are modest.

Importantly, `jina-v3`, despite being the lowest-dimensional model (1024-dim) and fully local, achieves competitive results—only 2-3 points below the best performing models on most metrics. This demonstrates that QAFD-RAG's query-aware flow diffusion mechanism is robust to embedding variations and can operate effectively with resource-constrained or privacy-preserving local embeddings. The consistent performance across diverse architectures (cloud vs. local, 1024-dim to 4096-dim) validates that our theoretical framework successfully leverages semantic similarity regardless of the specific embedding space.

### A2.1.5  LONG-DOCUMENT SUMMARIZATION: SQUALITY EVALUATION DETAILS

We evaluate on all 250 questions from the SQUALITY training dataset (Wang et al., 2022), which contains narrative passages paired with comprehension questions and multiple human-written reference answers per question. We compute BLEU-1 and BLEU-2 using unigram and bigram modified

Table 7: Embedding sensitivity analysis: QAFD-RAG performance across five embedding models on the Mix dataset. All numbers are mean (± stdev) over 5 runs.

| Embedding Model | Comprehensive. | Diversity | Logicality | Relevance | Coherence |
|---|---|---|---|---|---|
| text-embedding-3-small (1536d) | 87.15 (±3.46) | 81.15 (±4.86) | 90.70 (±2.93) | 94.36 (±4.50) | 90.36 (±2.07) |
| text-embedding-3-large (3072d) | **88.12 (±3.32)** | **83.08 (±4.71)** | 91.28 (±2.85) | **95.12 (±4.35)** | 90.85 (±2.01) |
| jina-v3 (1024d) | 85.82 (±3.58) | 80.18 (±5.02) | 89.18 (±3.05) | 91.98 (±4.68) | 87.94 (±2.15) |
| NVIDIA-nv-embed-v2 (4096d) | 87.85 (±3.41) | 82.45 (±4.78) | **91.58 (±2.81)** | 94.78 (±4.42) | **91.45 (±1.98)** |
| GritLM-7B (4096d) | 87.32 (±3.51) | 81.64 (±4.91) | 89.82 (±2.97) | 93.52 (±4.55) | 90.28 (±2.10) |

precisions with geometric averaging. BLEU-$n$ is defined as

$$\text{BLEU-}n := \exp\left(\sum_{i=1}^{n} w_i \log p_i\right),$$

where $p_i$ is the $i$-gram precision and $(w_1, \ldots, w_n)$ are the weights. For BLEU-1 we use $(1, 0, 0, 0)$; for BLEU-2 we use $(0.5, 0.5, 0, 0)$. ROUGE-1 F1 and ROUGE-2 F1 report F1 overlap of unigrams and bigrams using Porter stemming.

METEOR combines unigram precision (P) and recall (R) with a fragmentation penalty:

$$\text{METEOR} := F_{\text{mean}} \cdot \left(1 - \gamma \cdot \left(\frac{\text{chunks}}{\text{matches}}\right)^{\beta}\right),$$

where harmonic mean $F_{\text{mean}} := P \cdot R / (\alpha P + (1 - \alpha)R)$, chunks denotes contiguous matched segments and matches denotes total matched unigrams. We use Natural Language Toolkit (NLTK) defaults: $\alpha = 0.9$, $\beta = 3.0$, and $\gamma = 0.5$. All metrics use lowercased, word-tokenized text with BLEU/METEOR computed against all references.

A knowledge graph is constructed once from all narrative documents and reused across queries. All experiments use hybrid retrieval mode with diffusion parameter $\alpha = 10$, a 40-node source budget, and minimum flow threshold 0.01. Answers are produced with GPT-4o-mini using greedy decoding and full caching.

## A2.2 TEXT-TO-SQL

### A2.2.1 HYPERPARAMETER SETTINGS

We report the full set of QAFD-RAG hyperparameters used in our implementation. The most critical parameters are the *source mass* and the *target capacity*. The source mass is adaptively set based on the degree of the source node, following local clustering strategies in Fountoulakis et al. (2020b). Specifically, we define $m_s = \alpha \sum_{j \in P} T_j$, where $P$ is the set of nodes on the shortest path from source to target. The multiplier $\alpha = 10$ controls the initial injected mass; larger values ensure sufficient propagation across the graph. If no path exists between source and target, we apply a fallback rule: $m_s = \sum_{i \in \mathcal{V}} T_i$. Each node, including the target, is assigned a uniform sink capacity $T_i = 1$ and initialized with zero mass. During diffusion, nodes absorb incoming flow up to their capacity, with excess redistributed until convergence.

Other hyperparameters regulate the iterative behavior of the algorithm, to which it is generally robust. The convergence threshold in Algorithm 2 is set to $\epsilon = 0.05$, terminating when total excess mass falls below this threshold. To safeguard against non-convergence, we cap iterations at $N_{\text{max}} = 10^6$ and check convergence every 100 steps. For stability during edge reweighting, a constant $10^{-10}$ is added to all computations.

### A2.2.2 EXAMPLES OF SCHEMA IDENTIFIED BY QAFD-RAG FOR SPIDER2'S QUERIES

The core efficiency of QAFD-RAG stems from its ability to globally reason over the schema graph through flow diffusion, directly identifying the most relevant multi-hop paths between schema elements implicated by the query. As illustrated in Figure 4, QAFD-RAG, unlike Spider-Agent and other ReAct-style methods—which must sequentially probe the schema, issuing a separate LLM call

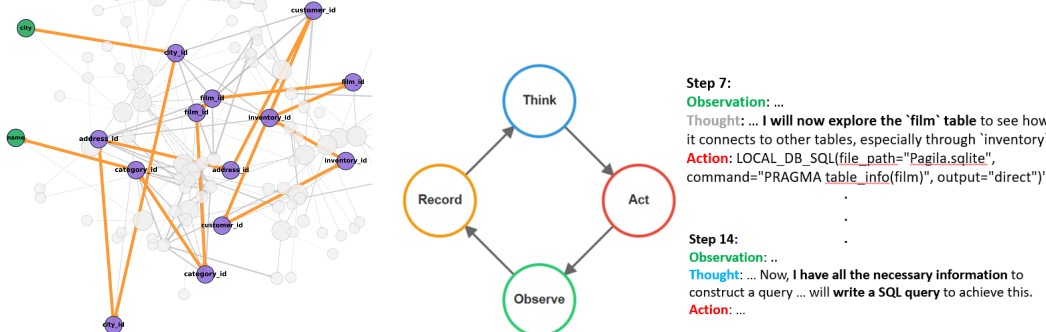

Figure 4: **Left:** *Schema path identified by QAFD-RAG for SQL generation on query* `local039` *(Lei et al., 2025).* **Right:** *Stepwise schema exploration for the same query using Spider-Agent (ReAct) (Lei et al., 2025).* QAFD-RAG discovers the full reasoning path in one pass, sharply reducing LLM calls. In contrast, Spider-Agent incrementally explores the schema, requiring 14 calls and higher latency. This comparison highlights QAFD-RAG's efficiency and accuracy on complex multi-hop queries.

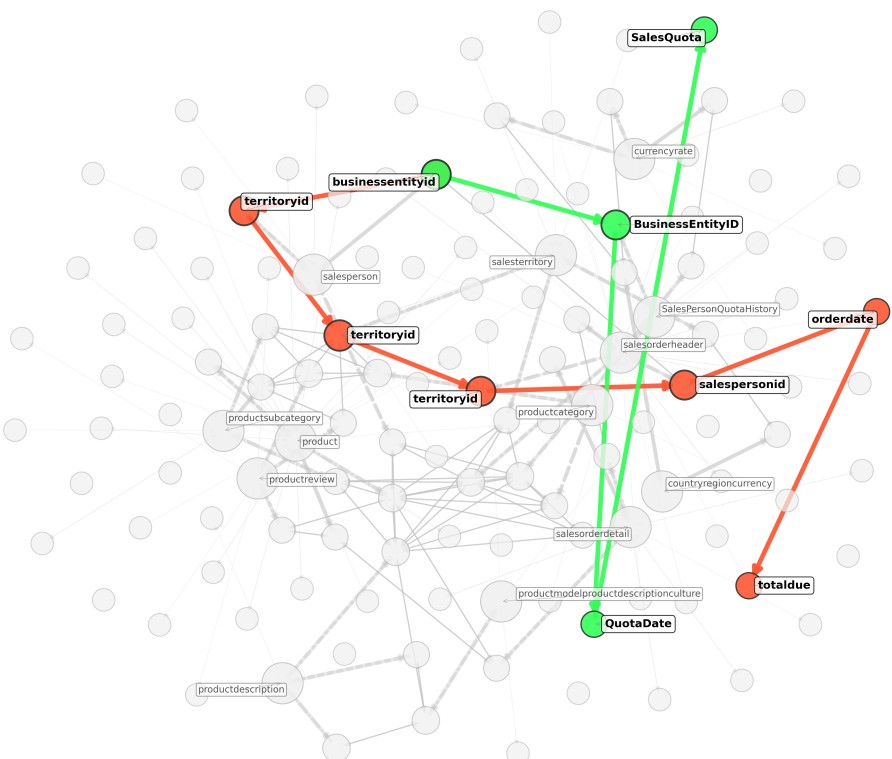

Figure 5: QAFD-RAG retrieved schema for Query-Local141 in the AdventureWorks database KG (Lei et al., 2025).

for each reasoning step, table, or join—performs a single, structured optimization to simultaneously discover all semantically relevant paths. This eliminates redundant exploration and dramatically reduces both the number of LLM calls and overall inference time, especially for queries requiring long or complex join paths. Consequently, QAFD-RAG delivers scalable inference for large enterprise databases while maintaining or even improving accuracy.

We provide an additional example to illustrate the advantages of our framework in uncovering meaningful schema paths, using Query-Local141 from the AdventureWorks database (Lei et al., 2025).

Table 8: LLM call efficiency across SQLite and Snowflake. Lower is better.

| Method | SQLite LLM Calls | Snowflake LLM Calls |
|---|---|---|
| Spider-Agent | 724 | 1482 |
| **QAFD-RAG + SQL-Agent** | **493** | **674** |

Table 9: Schema linking performance (precision, recall, F1). Higher is better.

| Method | SQLite | | | Snowflake | | |
|---|---|---|---|---|---|---|
| | **Prec** | **Rec** | **F1** | **Prec** | **Rec** | **F1** |
| Spider-Agent | 0.81 | 0.75 | 0.78 | 0.35 | 0.35 | 0.35 |
| **QAFD-RAG** | **0.82** | **0.76** | **0.79** | **0.60** | **0.59** | **0.60** |

**Example 1** (**Query-Local141**). *How did each salesperson's annual total sales compare to their annual sales quota? Provide the difference between their total sales and the quota for each year, organized by salesperson and year*

Figure 5 visualizes the relevant portion of the complex database schema as a graph, highlighting two distinct schema path decompositions: one corresponding to the sales quota (green path) and the other to the total sales (orange path). These paths not only capture the necessary schema linking but also reveal the underlying reasoning required to answer the query, as seen from the traversal across key nodes such as SalesQuota, totaldue, orderdate, and identifiers such as BusinessEntityID and salespersonid.

While Spider-Agent fails on this complex multi-hop reasoning query, QAFD-RAG effectively leverages the discovered paths to solve the problem, demonstrating the power of path-based schema linking and reasoning for compositional SQL logic.

### A2.2.3 LLM EFFICIENCY AND SCHEMA LINKING PERFORMANCE

Table 8 demonstrates that QAFD-RAG + SQL-Agent significantly reduces LLM call overhead, using 31.9% fewer LLM calls on SQLite and 54.5% fewer calls on Snowflake compared to the standard Spider-Agent approach. Overall, QAFD-RAG + SQL-Agent consistently outperforms baselines across both environments.

Finally, Table 9 presents detailed schema linking performance measured by precision, recall, and F1-score. It demonstrates the core advantage of QAFD-RAG in identifying relevant schema elements (tables and columns) for the given queries, explaining both its accuracy improvements and efficiency gains over Spider-Agent.

### A2.2.4 EMBEDDING MODEL SENSITIVITY FOR TEXT-TO-SQL

To evaluate QAFD-RAG's robustness to embedding quality in the Text-to-SQL domain, we assess schema linking performance across five diverse embedding models on the Local category (SQLite databases). We compare: (i) OpenAI's text-embedding-3-small (1536-dim) and text-embedding-3-large (3072-dim), representing cloud-based proprietary embeddings; (ii) Jina AI's jina-embeddings-v3 (Sturua et al., 2024) (1024-dim), an open-source local model; (iii) NVIDIA's nv-embed-v2 (Lee et al., 2024) (4096-dim), optimized for retrieval tasks; and (iv) GritLM-7B (Muennighoff et al., 2024) (4096-dim), a unified generative-embedding model. All other hyperparameters remain fixed across experiments.

Table 10 presents schema linking results. QAFD-RAG demonstrates consistent performance across all embedding models, with precision scores ranging from 0.80 to 0.83, recall from 0.76 to 0.81, and F1 scores from 0.78 to 0.82. Notably, nv-embed-v2 achieves the best performance across all metrics (0.83/0.81/0.82), which is specifically optimized for retrieval tasks, followed closely by text-embedding-3-large (0.82/0.77/0.79) and GritLM (0.82/0.78/0.80).

Table 10: Embedding sensitivity analysis for Text-to-SQL (Local category): QAFD-RAG performance across five embedding models.

| Embedding Model | Precision | Recall | F1-Score |
|---|---|---|---|
| text-embedding-3-small (1536d) | 0.82 | 0.76 | 0.79 |
| text-embedding-3-large (3072d) | 0.82 | 0.77 | 0.79 |
| jina-v3 (1024d) | 0.80 | 0.76 | 0.78 |
| NVIDIA-nv-embed-v2 (4096d) | **0.83** | **0.81** | **0.82** |
| GritLM-7B (4096d) | 0.82 | 0.78 | 0.80 |

Importantly, `jina-v3`, despite being the lowest-dimensional model (1024-dim) and fully local, achieves competitive results—only 0.04 points below the best model in F1 score. This demonstrates that QAFD-RAG's query-aware flow diffusion mechanism is robust to embedding variations and can operate effectively with resource-constrained or privacy-preserving local embeddings. The consistent performance across diverse architectures (cloud vs. local, 1024-dim to 4096-dim, ranging from 0.78 to 0.82 F1) validates that our framework successfully leverages semantic similarity for schema linking regardless of the specific embedding space, making it practical for deployment in varied infrastructure environments.

## A3    PROOF OF MAIN RESULTS

### A3.1    PROOF OF LEMMA 2

*Proof.* The proof follows from the structure of the push-relabel algorithm and properties of the dual formulation.

Algorithm 2 only increases $x_u$ when node $u$ has excess mass $m_u > T_u$. By the complementary slackness conditions of the dual problem (7), if $x_u^{(k)*} = 0$ at the optimum, then the corresponding constraint is inactive, meaning $m_u \leq T_u$ at optimality. Since the algorithm maintains the invariant that mass can only flow from nodes with excess to their neighbors, and the algorithm starts with $\mathbf{x}^0 = \mathbf{0}$, any node $u$ with $x_u^{(k)*} = 0$ will never have excess mass during the algorithm's execution. Therefore, $x_u^t = 0$ for all iterations $t$, proving $\mathrm{supp}(\mathbf{x}^t) \subseteq \mathrm{supp}(\mathbf{x}^{(k)*})$.

The source mass $\boldsymbol{\Delta}^{(k)}$ determines the total amount of "flow" injected into the system. Each unit of flow that reaches a node $u$ contributes at least a minimum amount to $x_u$ (bounded by the algorithm's push operations). Since mass is conserved and each node in the support must receive some positive flow to have $x_u > 0$, the number of nodes that can be in the support is upper bounded by the total mass $\|\boldsymbol{\Delta}^{(k)}\|_1$. Formally, if $|\mathrm{supp}(\mathbf{x}^{(k)*})| > \|\boldsymbol{\Delta}^{(k)}\|_1$, then the average flow per supported node would be less than 1, but the discrete nature of the push operations and capacity constraints ensure each supported node receives at least a minimum quantum of flow, leading to a contradiction.

$\square$

### A3.2    PROOF OF THEOREM 3

*Proof.* The proof follows the framework of coordinate descent analysis for flow diffusion problems (Fountoulakis et al., 2020b), adapted to handle query-aware edge weights. We analyze the expected decrease in the objective function at each iteration.

For simplicity, let $F(\mathbf{x}) = \frac{1}{2}\mathbf{x}^T\mathbf{L}(q_k)\mathbf{x} + \mathbf{x}^T(\mathbf{T} - \boldsymbol{\Delta}^{(k)})$ be the dual objective from (7), where $\mathbf{L}(q_k) = \mathbf{B}^T\bar{\mathbf{W}}(q_k)\mathbf{B}$ is the query-aware weighted Laplacian. At iteration $t$, Algorithm 2 selects a node $u$ with excess mass $m_u > T_u$ uniformly at random and performs a push operation. The mass vector maintains the relationship $\mathbf{m} = \boldsymbol{\Delta}^{(k)} - \mathbf{L}(q_k)\mathbf{x}$, so the gradient is $\nabla F(\mathbf{x}) = \mathbf{L}(q_k)\mathbf{x} + (\mathbf{T} - \boldsymbol{\Delta}^{(k)}) = \mathbf{T} - \mathbf{m}$. When node $u$ is selected, the excess mass is excess $= m_u - T_u > 0$, which corresponds to $\frac{\partial F}{\partial x_u} = T_u - m_u < 0$. The algorithm increases $x_u$ by $\frac{\text{excess}}{w_u}$ where $w_u = \sum_{v \sim u} \bar{w}(q_k, u, v)$, and redistributes the excess to neighbors.

The key insight is that this corresponds to a coordinate descent step. The expected decrease in the objective function is

$$\mathbb{E}[F(\mathbf{x}^{t+1}) - F(\mathbf{x}^t)] = -\mathbb{E}\left[\frac{(\text{excess})^2}{2w_u}\right] \cdot \mathbb{P}(\text{node } u \text{ is selected}) \tag{10}$$

Since nodes are selected uniformly from the excess set, and by Lemma 2, all iterates have support contained in $\text{supp}(\mathbf{x}^{(k)*})$, we have

$$\mathbb{E}[F(\mathbf{x}^{t+1}) - F(\mathbf{x}^t)] \leq -\frac{1}{|\text{supp}(\mathbf{x}^{(k)*})|} \cdot \frac{\eta^{(k)}}{2\gamma^{(k)}} \cdot \|\mathbf{T} - \mathbf{m}^t\|_1^2 \tag{11}$$

where we use $\eta^{(k)} \leq \bar{w}(q_k, u, v)$ for edges in the optimal support (minimum weight); $\sum_{v \sim u} \bar{w}(q_k, u, v) \leq \gamma^{(k)}$ for nodes in the optimal support (maximum weighted degree); and The excess mass at each node is bounded by the gradient components.

The total excess mass is $\|\mathbf{T} - \mathbf{m}^t\|_1$, and by the optimality conditions, this relates to the suboptimality as

$$F(\mathbf{x}^t) - F(\mathbf{x}^{(k)*}) \leq C \cdot \|\mathbf{T} - \mathbf{m}^t\|_1 \tag{12}$$

for some constant $C$ depending on the problem parameters. Combining these bounds and using the fact that $|\text{supp}(\mathbf{x}^{(k)*})| \leq \|\mathbf{\Delta}^{(k)}\|_1$ from Lemma 2, we get

$$\mathbb{E}[F(\mathbf{x}^{t+1}) - F(\mathbf{x}^{(k)*})] \leq \left(1 - \frac{\eta^{(k)}}{C \cdot \|\mathbf{\Delta}^{(k)}\|_1 \cdot \gamma^{(k)}}\right) \mathbb{E}[F(\mathbf{x}^t) - F(\mathbf{x}^{(k)*})] \tag{13}$$

This gives exponential convergence with rate $\frac{\eta^{(k)}}{\|\mathbf{\Delta}^{(k)}\|_1 \cdot \gamma^{(k)}}$. To achieve $\mathbb{E}[F(\mathbf{x}^{\tau^{(k)}})] - F(\mathbf{x}^{(k)*}) \leq \xi$, we need

$$\tau^{(k)} = O\left(\|\mathbf{\Delta}^{(k)}\|_1 \frac{\gamma^{(k)}}{\eta^{(k)}} \log \frac{1}{\xi}\right). \tag{14}$$

The query-aware nature enters through the weights $\bar{w}(q_k, u, v)$ in the definitions of $\gamma^{(k)}$ and $\eta^{(k)}$, meaning the convergence rate adapts to the semantic structure induced by the query, while maintaining the same algorithmic guarantees as classical flow diffusion. $\qquad \square$

### A3.3 PROOF OF LEMMA 6

*Proof.* Note that for any nodes $u, v \in \mathcal{V}$ and $k \in [K]$, we have

$$\|\mathbf{e}_u - \mathbf{e}_{q_k}\|^2 = \|\boldsymbol{\mu}_u - \boldsymbol{\mu}_{q_k}\|^2 + \|\mathbf{z}_u - \mathbf{z}_{q_k}\|^2 + 2\langle\boldsymbol{\mu}_u - \boldsymbol{\mu}_{q_k}, \mathbf{z}_u - \mathbf{z}_{q_k}\rangle, \tag{15a}$$

$$\|\mathbf{e}_u - \mathbf{e}_v\|^2 = \|\boldsymbol{\mu}_u - \boldsymbol{\mu}_v\|^2 + \|\mathbf{z}_u - \mathbf{z}_v\|^2 + 2\langle\boldsymbol{\mu}_u - \boldsymbol{\mu}_v, \mathbf{z}_u - \mathbf{z}_v\rangle. \tag{15b}$$

To analyze the concentration of $\|\mathbf{e}_u - \mathbf{e}_{q_k}\|^2$, note that $\|\mathbf{z}_u - \mathbf{z}_{q_k}\|^2 = \sum_{\ell=1}^d (z_{u\ell} - z_{q_k\ell})^2$. Each term $(z_{u\ell} - z_{q_k\ell})^2 - \mathbb{E}[(z_{u\ell} - z_{q_k\ell})^2]$ is sub-exponential with parameter at most $C\sigma_\ell^2$ for some absolute constant $C$ (see, e.g., (Vershynin, 2018, Theorem 2.7.7)). Applying Bernstein's inequality for sums of independent sub-exponential random variables and setting $t = C_1 \hat{\sigma}^2 \log |\mathcal{V}|$, we obtain

$$\|\mathbf{z}_u - \mathbf{z}_{q_k}\|^2 \leq 2\sum_{\ell=1}^d \sigma_\ell^2 + C_1 \hat{\sigma}^2 \log |\mathcal{V}| \tag{16a}$$

with probability at least $1 - 2|\mathcal{V}|^{-c'}$, where $\hat{\sigma} = \max_{1 \leq \ell \leq d} \sigma_\ell$ and $c' > 0$ is a constant.

For the cross term, observe that

$$\langle\boldsymbol{\mu}_u - \boldsymbol{\mu}_{q_k}, \mathbf{z}_u - \mathbf{z}_{q_k}\rangle = \sum_{\ell=1}^d (\mu_{u\ell} - \mu_{q_k\ell})(z_{u\ell} - z_{q_k\ell})$$

is a sum of independent, mean-zero sub-Gaussian random variables. By standard sub-Gaussian tail bounds (e.g., Hoeffding's inequality), for any $C_2 > 0$,

$$\mathbb{P}\left(\left|\langle\boldsymbol{\mu}_u - \boldsymbol{\mu}_{q_k}, \mathbf{z}_u - \mathbf{z}_{q_k}\rangle\right| > C_2 \hat{\sigma} \sqrt{\log |\mathcal{V}|} \|\boldsymbol{\mu}_u - \boldsymbol{\mu}_{q_k}\|\right) \leq 2|\mathcal{V}|^{-c''}$$

for some constant $c'' > 0$. Hence,

$$\left|\langle \boldsymbol{\mu}_u - \boldsymbol{\mu}_{q_k}, \mathbf{z}_u - \mathbf{z}_{q_k}\rangle\right| \leq C_2 \hat{\sigma} \sqrt{\log |\mathcal{V}|} \|\boldsymbol{\mu}_u - \boldsymbol{\mu}_{q_k}\| \tag{16b}$$

with probability at least $1 - 2|\mathcal{V}|^{-c''}$.

Therefore, with high probability,

$$\|\mathbf{e}_u - \mathbf{e}_{q_k}\|^2 \leq \|\boldsymbol{\mu}_u - \boldsymbol{\mu}_{q_k}\|^2 + 2\sum_{\ell=1}^{d} \sigma_\ell^2 + C_1 \hat{\sigma}^2 \log |\mathcal{V}|$$
$$+ 2C_2 \hat{\sigma} \sqrt{\log |\mathcal{V}|} \|\boldsymbol{\mu}_u - \boldsymbol{\mu}_{q_k}\|.$$

Following a similar argument, we derive bounds for $\|\mathbf{z}_u - \mathbf{z}_v\|^2$ and $\langle \boldsymbol{\mu}_u - \boldsymbol{\mu}_v, \mathbf{z}_u - \mathbf{z}_v\rangle$. Thus, for $u, v \in \mathcal{R}_k$, since $\boldsymbol{\mu}_u = \boldsymbol{\mu}_v = \boldsymbol{\mu}_{q_k}$, the decomposition in (15) simplifies, and we obtain, with probability at least $1 - O(|\mathcal{V}|^{-2})$

$$\bar{w}(q_k, u, v) \geq w(u, v) \exp\left(-(\gamma_1 + \gamma_2 + \gamma_3) \cdot O(\hat{\sigma}^2 d)\right)$$
$$= w(u, v) \exp(-o(1)) = w(u, v)(1 - o(1)).$$

This proves Item (i) in the lemma.

On the other hand, if $u \in \mathcal{R}_k$ and $v \in \mathcal{V} \setminus \mathcal{R}_k$, we have $\min(\|\boldsymbol{\mu}_u - \boldsymbol{\mu}_v\|, \|\boldsymbol{\mu}_v - \boldsymbol{\mu}_{q_k}\|) \geq \hat{\mu}$ for some $\hat{\mu}$. Thus, with high probability

$$\|\mathbf{e}_v - \mathbf{e}_{q_k}\|^2 \geq \hat{\mu}^2 - 2C_2 \hat{\sigma} \sqrt{\log |\mathcal{V}|} \hat{\mu} - O(\hat{\sigma}^2 d)$$
$$\geq \hat{\mu}^2(1 - o(1)).$$

From Assumption A, we get

$$\bar{w}(q_k, u, v) \leq w(u, v) \exp(-\gamma_2 \hat{\mu}^2(1 - o(1))) = w(u, v) \exp(-\omega(\log |\mathcal{V}|)).$$

Since the results hold uniformly over all pairs of nodes $(u, v)$, applying a union bound over all $O(|\mathcal{V}|^2)$ edges completes the proof. $\square$

### A3.4 PROOF OF THEOREM 7

*Proof.* Consider Problem (5). Under Assumption A, the induced subgraph on $\mathcal{R}_k$ is connected. (Otherwise, the alternative probabilistic condition on $\rho_1$ guarantees expansion—and thus connectivity—with high probability by Chernoff bound arguments; see, e.g., (Yang & Fountoulakis, 2023, Lemma C.1).)

Order the nodes in $\mathcal{R}_k$ as $v_1, v_2, \ldots, v_{r_k}$ such that $x_{v_1}^{(k)} \geq x_{v_2}^{(k)} \geq \cdots \geq x_{v_{r_k}}^{(k)}$, and let $x_{v_{r_k}}^{(k)} = 0$ be the minimum. As the subgraph on $\mathcal{R}_k$ is connected, there is a subgraph $(v_1 = u_0, \ldots, u_m = v_{r_k})$ with $m \leq r_k - 1$, where each $(u_\ell, u_{\ell+1})$ is an edge within $\mathcal{R}_k$. By the KKT conditions, for every edge $(u, v)$ with $x_u^{(k)} > x_v^{(k)}$, the optimal solution must satisfy

$$\bar{w}(q_k, u, v)(x_u^{(k)} - x_v^{(k)}) \leq (1 + \beta) \sum_{u \in \mathcal{R}_k} T_u.$$

Therefore, along each edge of the subgraph

$$x_{u_\ell}^{(k)} - x_{u_{\ell+1}}^{(k)} \leq \frac{(1 + \beta)}{\bar{w}(q_k, u_\ell, u_{\ell+1})} \sum_{u \in \mathcal{R}_k} T_u$$

for $0 \leq \ell \leq m - 1$.

By the edge separation property in Lemma 6, $\bar{w}(q_k, u_\ell, u_{\ell+1}) \geq w(u_\ell, u_{\ell+1})(1 - o(1))$ for intra-$\mathcal{R}_k$ edge. Summing the above along the path, we have

$$x_{v_1}^{(k)} = \sum_{\ell=0}^{m-1} (x_{u_\ell}^{(k)} - x_{u_{\ell+1}}^{(k)}) + x_{v_{r_k}}^{(k)} \leq \sum_{\ell=0}^{m-1} \frac{(1 + \beta) \sum_{u \in \mathcal{R}_k} T_u}{1 - o(1)}$$
$$\leq \frac{r_k(1 + \beta) \sum_{u \in \mathcal{R}_k} T_u}{1 - o(1)}. \tag{17}$$

The total mass injected at sources in $\mathcal{R}_k$ is $(1 + \beta) \sum_{u \in \mathcal{R}_k} T_u$. Nodes in $\mathcal{R}_k$ can absorb at most $\sum_{u \in \mathcal{R}_k} T_u$ mass. Therefore, at least $\beta \sum_{u \in \mathcal{R}_k} T_u$ mass must flow out of $\mathcal{R}_k$.

Next, we provide upper bound on the outflow. The total flow leaving $\mathcal{R}_k$ is

$$\text{Outflow} := \sum_{u \in \mathcal{R}_k} \sum_{\substack{v \notin \mathcal{R}_k \\ (u,v) \in \mathcal{E}}} \bar{w}(q_k, u, v)(x_u^{(k)} - x_v^{(k)})^+ \leq \sum_{u \in \mathcal{R}_k} x_u^{(k)} \sum_{\substack{v \notin \mathcal{R}_k \\ (u,v) \in \mathcal{E}}} \bar{w}(q_k, u, v)$$

$$\leq x_{v_1}^{(k)} \sum_{u \in \mathcal{R}_k} \sum_{\substack{v \notin \mathcal{R}_k \\ (u,v) \in \mathcal{E}}} \bar{w}(q_k, u, v). \tag{18}$$

By Lemma 6, $\bar{w}(q_k, u, v) \leq \exp(-\omega(\log|\mathcal{V}|))$ for $u \in \mathcal{R}_k, v \notin \mathcal{R}_k$. Let $N_{out}$ be the number of edges from $\mathcal{R}_k$ to $\mathcal{V} \setminus \mathcal{R}_k$. By Chernoff bounds, we have

$$\text{Prob}(N_{out} > 2\rho_2 r_k(|\mathcal{V}| - r_k)) \leq \exp(-\Omega(\rho_2 r_k(|\mathcal{V}| - r_k))). \tag{19}$$

Therefore, with high probability

$$\text{Outflow} \leq \frac{r_k(1 + \beta) \sum_{u \in \mathcal{R}_k} T_u}{1 - o(1)} \cdot 2\rho_2 r_k(|\mathcal{V}| - r_k) \cdot \exp(-\omega(\log|\mathcal{V}|))$$

$$= (1 + \beta) \sum_{u \in \mathcal{R}_k} T_u \cdot \text{poly}(|\mathcal{V}|) \cdot \exp(-\omega(\log|\mathcal{V}|))$$

$$= o\left(\beta \sum_{u \in \mathcal{R}_k} T_u\right).$$

This contradicts the requirement that at least $\beta \sum_{u \in \mathcal{R}_k} T_u$ mass must leave $\mathcal{R}_k$. Thus, our assumption that $x_{v_{r_k}}^{(k)} = 0$ is false, so $x_v^{(k)} > 0$ for all $v \in \mathcal{R}_k$, i.e., $\mathcal{R}_k \subseteq \text{supp}(\mathbf{x}^{(k)})$.

Finally, the mass that does leave $\mathcal{R}_k$ must be absorbed at nodes outside $\mathcal{R}_k$. Since each $x_u^{(k)} \leq T_u$ (by the constraints), the sum of $T_u$ over nodes in $\text{supp}(\mathbf{x}^{(k)}) \setminus \mathcal{R}_k$ cannot exceed the total outflow, i.e.,

$$\sum_{u \in \text{supp}(\mathbf{x}^{(k)}) \setminus \mathcal{R}_k} T_u \leq \beta \sum_{u \in \mathcal{R}_k} T_u.$$

This establishes the second part of the theorem. The desired high-probability result then follows by a union bound. □

## A4  OVERVIEW OF DATASETS AND PROMPTS USED IN QAFD-RAG FOR QUESTION ANSWERING

This section provides the details of datasets and key prompts used throughout our framework. To ensure fair comparison and eliminate confounding effects due to prompt engineering, we directly adopt the prompt templates introduced in Chen et al. (2025a) and Guo et al. (2024) for multiple stages of our pipeline. Specifically, the prompts for knowledge graph indexing, keyword extraction, and RAG-based query answering are reused without modification. For the performance evaluation stage, we also follow Chen et al. (2025a) and Guo et al. (2024)'s evaluation setup by prompting the model to rate responses along five predefined dimensions: Comprehensiveness, Diversity, Logicality, Relevance, and Coherence. These standardized definitions and formulations are explicitly included in our evaluation prompts. Using a consistent prompt set across all models ensures the reliability and reproducibility of our experimental results.

### A4.1  DATASETS

### A4.2  ENTITY AND RELATIONSHIP EXTRACTION PROMPT

This prompt is used to extract structured entity and relation information from individual document chunks, forming the nodes and edges of the knowledge graph. The prompt format is directly adopted from Chen et al. (2025a).

Table 11: Dataset statistics for the UltraDomain subsets used in our evaluation.

| Dataset | # of documents | # of tokens | # of nodes in the indexing graph | # of questions |
|---------|---------------|-------------|----------------------------------|----------------|
| Agriculture | 12 | 1,949,526 | 23,180 | 100 |
| Biology | 27 | 3,275,990 | 42,520 | 220 |
| Cooking | 14 | 2,232,441 | 18,985 | 120 |
| History | 26 | 5,159,599 | 63,840 | 180 |
| Legal | 94 | 4,773,793 | 20,838 | 438 |
| Mathematics | 20 | 3,640,908 | 32,319 | 160 |
| Mix | 61 | 611,161 | 11,371 | 130 |
| Music | 29 | 5,038,910 | 58,245 | 200 |
| Philosophy | 26 | 3,561,642 | 33,241 | 200 |
| Physics | 19 | 2,116,825 | 19,745 | 160 |

## Prompt 1: Entity and Relationship Extraction

—Goal—

Given a text document that is potentially relevant to this activity and a list of entity types, identify all entities of those types from the text and all relationships among the identified entities. Use {language} as output language.

—Steps—

1. Identify all entities. For each identified entity, extract the following information:

   - entity_name: Name of the entity, use the same language as input text. If English, capitalize the name.
   - entity_type: One of the following types: [{entity_types}]
   - entity_description: Comprehensive description of the entity's attributes and activities

   Format each entity as
   ("entity"{tuple_delimiter}<entity_name>{tuple_delimiter}
   <entity_type>{tuple_delimiter}<entity_description>)

2. From the entities identified in step 1, identify all pairs of (source_entity, target_entity) that are clearly related to each other. For each pair of related entities, extract the following information:

   - source_entity: name of the source entity, as identified in step 1
   - target_entity: name of the target entity, as identified in step 1
   - relationship_description: explanation as to why you think the source entity and the target entity are related to each other
   - relationship_strength: a numeric score indicating the strength of the relationship between the source entity and target entity
   - relationship_keywords: one or more high-level key words that summarize the overarching nature of the relationship, focusing on concepts or themes rather than specific details

   Format each relationship as
   ("relationship"{tuple_delimiter}<source_entity>{tuple_delimiter}<target_entity>
   {tuple_delimiter}<relationship_description>{tuple_delimiter}
   <relationship_keywords>{tuple_delimiter}<relationship_strength>)

3. Identify high-level keywords that summarize the main concepts, themes, or topics of the entire text. These should capture the overarching ideas present in the document. Format the content-level keywords as

   ("content_keywords"{tuple_delimiter}<high_level_keywords>)

4. Return output in {language} as a single list of all the entities and relationships identified in steps 1 and 2. Use **{record_delimiter}** as the list delimiter.

> 5. When finished, output {completion_delimiter}

### A4.3  KEYWORD EXTRACTION PROMPT

This prompt identifies salient concepts and entities from a given query for the purpose of initializing seed nodes during query-aware diffusion. It follows Chen et al. (2025a)'s design without any modification.

---

**Prompt 2: Keyword-Extraction in Knowledge Graph Question Answering**

—Role—
You are a helpful assistant tasked with identifying both high-level and low-level keywords in the user's query.

—Goal—
Given the query, list both high-level and low-level keywords. High-level keywords focus on overarching concepts or themes, while low-level keywords focus on specific entities, details, or concrete terms.

—Instructions—
- Output the keywords in JSON format.
- The JSON should have two keys:

    - "high_level_keywords" for overarching concepts or themes.
    - "low_level_keywords" for specific entities or details.

---

### A4.4  QUERY ANSWERING PROMPT

This prompt is used to generate the final response from the LLM based on retrieved evidence. We adopt Chen et al. (2025a)'s instruction format for fairness in model comparison.

---

**Prompt 3: RAG-based query answering**

—Role—
You are a helpful assistant responding to questions about the data in the tables provided.

—Goal—
Generate a response of the target length and format that responds to the user's question, summarizing all information in the input data tables appropriate for the response length and format, and incorporating any relevant general knowledge. If you don't know the answer, just say so. Do not make anything up. Do not include information where the supporting evidence for it is not provided.

—Target response length and format—
    {response_type}

—Data tables—
    {context_data}

Add sections and commentary to the response as appropriate for the length and format. Style the response in markdown.

---

### A4.5 EVALUATION PROMPT

This prompt is used to obtain ratings from the LLM across five dimensions: Comprehensiveness, Diversity, Logicality, Relevance, and Coherence. The full dimension definitions are embedded in the prompt and align with those defined in Chen et al. (2025a).

---

**Prompt 4: Performance Evaluation**

—Role—
You are an expert tasked with evaluating question answering based on five criteria: Comprehensiveness, Diversity, Logicality, Relevance, and Coherence.

—Goal—
Evaluate the following response to a question based on five criteria. Rate each criterion from 0 to 100.

Question: {query}
Response: {response}

Please evaluate based on these criteria:

- Comprehensiveness: How much detail does the answer provide to cover all aspects and details of the question?
- Diversity: How varied and rich is the answer in providing different perspectives and insights on the question?
- Logicality: How logically does the answer respond to all parts of the question?
- Relevance: How relevant is the answer to the question, staying focused and addressing the intended topic or issue?
- Coherence: How well does the answer maintain internal logical connections between its parts, ensuring a smooth and consistent structure?

Provide scores in JSON format:

```
{{
        "comprehensiveness": [score],
        "diversity": [score],
        "logicality": [score],
        "relevance": [score],
        "coherence": [score]
}}
```

---

## A5 OVERVIEW OF PROMPTS USED IN QAFD-RAG IN TEXT-TO-SQL TASKS

In the following, we provide a Text-to-SQL planning agent prompt to systematically analyze database graph nodes (tables and columns) in relation to user queries in order to identify graph seed nodes. The agent must decompose each query into subqueries, examine all schema nodes for semantic, structural, and temporal connections, and output a JSON object listing source-target (seed node candidate) pairs for each subquery. A similar prompt can be used for Snowflake graphs with minor modifications.

---

**Prompt 5: Seed Node Selection for SQLite Graph**

TASK DEFINITION

You are a Text-to-SQL planner agent.
Given:

---

- A user query: {QUERY}
- A database schema summary (graph) in text format: {SCHEMA_SUMMARY}

Your job is to:

1. Review this schema summary and user query thoroughly.

2. Break the user query into logical user subqueries.

3. **For each subquery, systematically examine EVERY single node (table.column) in the database schema graph one by one:**

   - Go through each table in the schema summary sequentially
   - For each table, examine every column within that table
   - For each table.column node, evaluate its relationship strength with the current subquery
   - Document your analysis for each node, determining if it has any of these RELATIONSHIP TYPES with the subquery:
     - SEMANTIC: Direct or indirect relevance to query concepts
     - STRUCTURAL: Representing organizational structures in the query
     - TEMPORAL: Time-based connections to query elements
     - CAUSAL: Cause-effect relationships described in the query
     - LOGICAL: Supporting logical conditions in the query
     - STATISTICAL: Statistical correlations to query concepts
     - DOMAIN-SPECIFIC: Domain relevancy with query
   - **Source nodes**: ALL STARTING COLUMNS (with table prefixes) having strong relationships of ANY TYPE ABOVE with the subquery.
   - **Target nodes**: ALL DESTINATION COLUMNS (with table prefixes) having strong relationships of ANY TYPE ABOVE with the subquery.
   - **MANDATORY**: You must examine and consider every single table.column combination in the schema before proceeding to the next step.

4. When domain-specific concepts appear in the query, properly map these concepts to the appropriate schema column elements

5. **Identify the most confident path of schema graph**: For each subquery, determine and explicitly state the most confident path that the LLM should follow through the schema graph, using right arrow format ($\rightarrow$) with ONLY schema nodes.

6. **Provide reasoning confidence candidates**: For each subquery, provide EXACTLY 2-3 DIVERSE candidates in format [source, target, confidence] where source is a starting node, target is an ending node, and confidence is a float between 0.0-1.0. Each candidate should explore different interpretations, relationship types, or alternative paths.

Important:

- Schema nodes MUST be specified as "table.column" EXACTLY the same as they appear in the schema summary. EVEN DO NOT change upper or lower letters.
- The most confident path should use the right arrow format ($\rightarrow$) and contain ONLY schema nodes (table.column format)
- Reasoning confidence candidates should be in format: [source, target, confidence] with EXACTLY 2-3 DIVERSE candidates per subquery
- Each candidate should represent different semantic interpretations, alternative join paths, or different relationship types
- **CRITICAL**: You MUST systematically go through every single table and every single column in the schema graph during step 3 analysis
- Only output a JSON without explanation

In the following, we provide an example of a Snowflake system prompt for SQL-Agent. A similar prompt was used for SQLite with minor modifications. The agent is required to systematically analyze database schemas and execute SQL queries by following a workflow that prioritizes the identification of the highest-reward subgraphs from the schema.json files, which were built using QAFD-RAG.

---

**Prompt 6: Snowflake Data Scientist Agent**

TASK DEFINITION

You are a data scientist proficient in database, SQL and DBT Project. You are starting in the {work_dir} directory, which contains all the data needed for your tasks. You can only use the actions provided in the ACTION SPACE to solve the task. For each step, you must output an Action; it cannot be empty. The maximum number of steps you can take is {max_steps}. Do not output an empty string!

ACTION SPACE

{action_space}

SNOWFLAKE-QUERY PROTOCOL

1. **BEFORE following any other rules, you MUST follow these steps:**
   - Read the schema.json in the /workspace directory. The schema.json contains valuable information about the graph structure and path rewards.
   - Identify the highest-reward subquery paths for EACH division in the provided schema.json
   - Write your VERY FIRST SQL query by combining ONLY these highest-reward paths to address the main query
   - DO NOT perform ANY database inspection before executing this first query
   - You MUST run this query as your FIRST SQL execution
   - You MUST terminate execution immediately after this first query if it works

   For your FIRST SQL attempt, follow ONLY these steps:
   - Review the schema.json structure containing all subqueries and their reward values
   - For each division (subquery), identify the path with the highest reward
   - Combine these highest-reward paths into a SINGLE comprehensive SQL query
   - Execute this SQL query immediately with no other database commands before it

   Do NOT run any exploratory queries like:
   - DO NOT run "SELECT * FROM table LIMIT 5"
   - DO NOT run "PRAGMA table_info(table_name)"
   - DO NOT run "SELECT name FROM sqlite_master WHERE type='table'"
   - DO NOT check data types
   - DO NOT check for NULL values
   - DO NOT try to understand the schema first
   - DO NOT check DDL.csv files before attempting the first query

   Your first query must be the direct SQL translation of combining all highest-reward paths to address the main query objective.

2. If your first query fails, then you should explore the database structure further using the methods below. You can check DDL.csv file with the database's DDL, along with JSON files that contain the column names, column types, column descriptions, and sample rows for individual tables. You can review the DDL.csv file in each directory, then selectively examine the JSON files as needed. Read them carefully.

3. You can use SNOWFLAKE_EXEC_SQL to run your SQL queries and interact with the database. Do not use this action to query INFORMATION_SCHEMA or SHOW DATABASES/TABLES; the schema information is all stored in the /workspace/-database_name folder. Refer to this folder whenever you have doubts about the schema.

---

4. Be prepared to write multiple SQL queries to find the correct answer. Once it makes sense, consider it resolved.

5. Focus on SQL queries rather than frequently using Bash commands like grep and cat, though they can be used when necessary.

6. If you encounter an SQL error, reconsider the database information and your previous queries, then adjust your SQL accordingly. Do not output the same SQL queries repeatedly.

7. Ensure you get valid results, not an empty file. Once the results are stored in result.csv, make sure the file contains data. If it is empty or just contains the table header, it means your SQL query is incorrect.

8. The final result MUST be a CSV file, not an .sql file, a calculation, an idea, a sentence or merely an intermediate step. Save the answer as a CSV and provide the file name, it is usually from the SQL execution result.

TIPS

1. When referencing table names in Snowflake SQL, you must include both the database_name and schema_name. For example, for
/workspace/DEPS_DEV_V1/DEPS_DEV_V1/ADVISORIES.json,
if you want to use it in SQL, you should write
DEPS_DEV_V1.DEPS_DEV_V1.ADVISORIES.

2. Do not write SQL queries to retrieve the schema; use the existing schema documents in the folders.

3. When encountering bugs, carefully analyze and think them through; avoid writing repetitive code.

4. Column names must be enclosed in quotes. But don't use \", just use ".

RESPONSE FORMAT

For each task input, your response should contain:

1. One analysis of the task and the current environment, reasoning to determine the next action (prefix "Thought: ").

2. One action string in the ACTION SPACE (prefix "Action: ").

EXAMPLE INTERACTION

Observation: ...(the output of last actions, as provided by the environment and the code output, you don't need to generate it) Thought: ... Action: ...

TASK

Please solve this task: {task}

## A6  TEXT-TO-SQL BASELINES

**CHASE-SQL for SQLite** We made several key modifications to adapt CHASE-SQL for the Spider 2.0 SQLite benchmark. Enhanced data preprocessing expanded the data types used for extracting unique values from TEXT-only to include all "CHAR"-containing types (CHAR, VARCHAR, NVAR-CHAR) present in Spider 2.0. We disabled schema selection comparison features to prevent failures when no golden SQL query is available, and updated candidate generation prompts to encourage SQL generation even in low-confidence scenarios, addressing the complexity-induced abstention issues in Spider 2.0.

Additional improvements included increased execution timeout thresholds to accommodate more complex SQL queries and enforcing a fixed agentic pipeline order: keyword extraction → entity

retrieval → context retrieval → column filtering → table selection → column selection → candidate generation → revision. This fixed order was necessary after observing that dynamic tool selection often led to suboptimal sequences.

**CHASE-SQL for Snowflake** All SQLite modifications were retained, but we disabled the Information Retrieval module, skipping entity and context retrieval phases. Transitioning to Snowflake required addressing three main challenges: syntax differences, database schema representation, and connection protocols.

While Snowflake SQL is largely compatible with SQLite, key differences include date format handling and using double quotes instead of backticks for quoted identifiers. The original CHESS schema representation using simple table-name dictionaries proved insufficient for Snowflake's hierarchical structure, where tables are organized into schemas within databases. We introduced schema-table mapping to handle this abstraction, though this assumes no duplicate table names across schemas.

Database connection also differs significantly. SQLite uses direct file paths, while Snowflake requires cloud account credentials with full namespace specification (database.schema.table) or reduced namespace (schema.table) when connected to a specific database. Golden SQL examples use full namespace format. Source code modifications addressed these data access differences. For prompt adaptation, we used the o3 model with instructions shown in Prompt 7 to modify generate_candidate_one and revise_one templates. Initial experiments revealed that full namespace format contradicted GPT-4o's internal knowledge, which prefers simplified schema.table format. We therefore modified prompts to use schema.table format while passing database names directly to the connector.

---

**Prompt 7:** **System prompt for o3 to adapt generate_candidate_one and revise_one prompts for snowflake dialect**

Help me modify this template to switch from SQLite dialect to snowflake dialect.
1. Make sure to change the syntax and functions specifically for snowflake, but do not change the prompt structure and do not omit any examples.
2. All examples must be executable in snowflake.
3. Make sure to change all ticks <'> with double quotes <">
4. All DDL statements specifying the database structure must include statements for creating a database and schema. You can infer the best names for the database and schema if they are not immediately available. Use the same name for the database and the schema. When referencing tables, use the full namespace including the database and schema, like so: database.schema.table

Here is an example:

CREATE DATABASE restaurants;
CREATE SCHEMA restaurants;
CREATE TABLE generalinfo
(
          id_restaurant INTEGER not null primary key,
          food_type TEXT null, – examples: 'thai'| 'food type' description: the food type
          city TEXT null, – description: the city where the restaurant is located in
);

CREATE TABLE location
(
          id_restaurant INTEGER not null primary key,
          street_name TEXT null, – examples: 'ave', 'san pablo ave', 'pablo ave'| 'street name'
description: the street name of the restaurant
          city TEXT null, – description: the city where the restaurant is located in
          foreign key (id_restaurant) references generalinfo (id_restaurant) on update cascade on
delete cascade,
);

---

> Use table location like so:
> restaurants.restaurants.location

Final prompt modifications included converting few-shot examples from SQLite to Snowflake syntax, adding Snowflake-specific instructions, implementing schema.table namespace usage, incorporating schema generation statements in DDL commands, and updating date processing instructions with Snowflake-specific functions and examples.

To establish strong baselines, we compared our approach against three additional methods from the Spider 2 repository: DIN-SQL (Pourreza & Rafiei, 2024), DAIL-SQL (Gao et al., 2023a), and CodeS (Li et al., 2024b). These methods represent different paradigmatic approaches to text-to-SQL generation.

**DIN-SQL: Decomposed In-Context Learning of Text-to-SQL with Self-Correction**   DIN-SQL (Pourreza & Rafiei, 2024) structures text-to-SQL generation through a four-stage decomposition approach. The method begins with schema linking to parse questions and identify references to database elements, followed by classification and decomposition that categorizes queries into easy (single-table), non-nested complex (joins without sub-queries), and nested complex (joins with sub-queries and set operations) classes. The SQL generation stage adapts its strategy to each class: simple few-shot prompting for easy queries, intermediate representations for non-nested complex queries, and sub-query decomposition with intermediate representations for nested complex cases. Finally, self-correction uses an LLM to identify and fix bugs in the generated SQL.

**DAIL-SQL: Dual-Similarity Adaptive In-Context Learning**   DAIL-SQL (Gao et al., 2023a) employs a five-stage pipeline centered on dual-similarity matching. The method starts with masking database-related tokens in both target and candidate questions, then uses a preliminary predictor for initial SQL prediction. Skeleton extraction identifies structural patterns in both predicted and candidate queries. The core innovation lies in sorting and reordering, where candidates are first sorted by masked question similarity and then reordered by prioritizing high query similarity matches. Finally, generation produces the final SQL using the optimally ordered examples.

**CodeS: Domain-Adaptive SQL-Centric Model**   CodeS (Li et al., 2024b) takes a pretraining-based approach with three main components. Incremental pre-training builds upon StarCoder using a specially curated SQL dataset. Database prompt construction creates context by selecting top-k relevant tables and columns using BM25 and LCS matching, incorporating representative values and metadata. Bi-directional augmentation expands few-shot examples through SQL-to-NL and NL-to-SQL data augmentation before supervised fine-tuning. The model can operate through either supervised fine-tuning or direct few-shot in-context learning.

**Spider-Agent: ReAct-style (Yao et al., 2023) Agentic Framework**   We evaluate the Spider-Agent baseline (Lei et al., 2024), which implements a ReAct-style (Yao et al., 2023) multi-step reasoning framework for Text-to-SQL, strictly following the original settings and publicly released code[2] without any modifications to model parameters, prompt structure, or evaluation scripts. Specifically, all hyperparameters were kept at their default values as provided by the authors: the LLM model is set to GPT-4o, the decoding temperature is set to 0.5, the nucleus sampling parameter (`top_p`) is 0.9, the maximum generation length (`max_tokens`) is 2500, the agent's step limit is 20, and the agent memory length is 25.

All pipelines were executed using Spider 2 benchmark scripts (Lei et al., 2024) that convert data to each method's preferred format. To maintain baseline authenticity while ensuring runnable outputs, we applied only minimal fixes necessary for execution, deliberately avoiding substantive tuning. Testing covered both Snowflake and Lite subsets of Spider 2.

The enterprise-scale schemas in Spider 2 presented significant challenges across all baseline methods. Context-length overflow occurred frequently as schemas exceeded token limits when serialized verbatim. Zero error-remediation was observed, where logical or syntactic failures (e.g., missing GROUP BY columns) were accepted without iterative correction or external validation. Additionally,

---

[2]https://github.com/xlang-ai/Spider2

malformatted data issues arose when benchmarks expected different data formats than those available in Spider 2, contributing to overall poor performance across methods.

## A7 USAGE OF LARGE LANGUAGE MODELS (LLMs)

We used large language models as a general-purpose writing assistant. Its role was limited to grammar checking, minor stylistic polishing, and improving the clarity of phrasing in some parts of the manuscript. The authors made all substantive contributions to the research and writing.

