# OpenReview forum: "Query-Aware Flow Diffusion for Graph-Based RAG with Retrieval Guarantees"
_ICLR.cc/2026/Conference — ICLR 2026 Poster_

### Official Review · Reviewer_Tki9 · 2025-10-30

**Soundness:** 3
**Presentation:** 4
**Contribution:** 3
**Rating:** 8
**Confidence:** 3

**Summary:**

This paper introduces Query-Aware Flow Diffusion (QAFD), a new graph-based retrieval method for retrieval-augmented generation. The key idea is to make graph diffusion sensitive to query semantics by dynamically adjusting edge weights based on semantic similarity. The authors also provide theoretical guarantees on convergence and subgraph recovery, and validate the method on multi-hop QA and text-to-SQL tasks, showing clear improvements over prior graph RAG approaches.

**Strengths:**

1. The proposed method is conceptually novel, integrating semantic and structural signals in a principled way.

2. Theoretical analysis provides valuable guarantees rarely seen in RAG research.

3. Experiments are comprehensive and show strong performance gains with reduced computation cost.

**Weaknesses:**

1. Despite the significant innovation in the graph construction process, the method still heavily relies on the quality of pretrained embeddings for semantic similarity, which may limit robustness under domain shift.

2. The method may struggle with handling logical negation. For instance, a query like “not red” may still activate nodes related to “red,” since it is unclear whether the embedding-based similarity explicitly encode negation or exclusion relationships. This could limit the model’s ability to capture fine-grained logical distinctions in retrieval.

**Questions:**

See weaknesses.

---

> ### Author Response · Authors · 2025-11-21
> **Response to Reviewer Tki9 – Part I**
>
> > **Weakness 1:** Despite the significant innovation ... relies on the quality of pretrained embeddings for semantic similarity, which may limit robustness under domain shift.
>
> **Response:** Thank you for your insightful comment. We agree that understanding QAFD-RAG's sensitivity to embedding quality—as discussed in the limitation section (Section 5)—is crucial for demonstrating cross-domain robustness.
>
> In the revision, we have added comprehensive embedding sensitivity experiments evaluating QAFD-RAG across five diverse embedding models with varying dimensions (1024–4096) and architectures: [OpenAI's text-embedding-3-small (1536-dim)](https://platform.openai.com/docs/guides/embeddings) and [text-embedding-3-large (3072-dim)](https://platform.openai.com/docs/guides/embeddings), [Jina AI's jina-v3 (1024-dim)](https://arxiv.org/abs/2409.10173), [NVIDIA's nv-embed-v2 (4096-dim)](https://arxiv.org/abs/2405.17428), and [GritLM-7B (4096-dim)](https://arxiv.org/abs/2402.09906). We evaluate on two distinct tasks: (1) question answering on UltraDomain Mix (Appendix A2.1.4), and (2) Text-to-SQL on SQLite (Appendix A2.2.4).
>
> QAFD-RAG demonstrates stable performance across all five models on both tasks, with Comprehensiveness
> ranging from 85.82 to 88.12 on UltraDomain and F1 scores from 0.78 to 0.82 on Text-to-SQL schema
> linking. Notably, jina-v3 (1024-dim), a local open-source model, remains competitive—only 2–3
> points below the best models on UltraDomain and only 0.04 F1 points below on Text-to-SQL. This
> demonstrates that QAFD-RAG's query-aware flow diffusion successfully leverages semantic similarity
> regardless of embedding architecture, and works effectively with resource-constrained or privacy-
> preserving local embeddings. Analysis across these two fundamentally different tasks (open-domain
> QA and structured schema linking) shows that QAFD-RAG maintains strong retrieval effectiveness
> across diverse embedding choices.
>
> All other experiments use text-embedding-3-small as the default (chosen for its balance of performance and efficiency), showing that QAFD-RAG achieves strong performance from an efficient baseline while maintaining robustness across diverse architectures.
>
> ---
>
> Following your feedback, we made the following revisions:
>
> ### Added to Appendix A2.1.4: Embedding Model Sensitivity Analysis (UltraDomain Mix)
>
> To assess QAFD-RAG's robustness to embedding quality, we evaluate the framework across five diverse embedding models on the Mix dataset from UltraDomain. We compare: (i) OpenAI's text-embedding-3-small (1536-dim) and text-embedding-3-large (3072-dim), representing cloud-based proprietary embeddings; (ii) Jina AI's jina-embeddings-v3 (1024-dim), an open-source local model; (iii) NVIDIA's nv-embed-v2 (4096-dim), optimized for retrieval tasks; and (iv) GritLM-7B (4096-dim), a unified generative-embedding model. All other hyperparameters remain fixed across experiments.
>
> Table 7 presents results across five evaluation dimensions. QAFD-RAG demonstrates consistent performance across all embedding models, with Comprehensiveness scores ranging from 85.82 to 88.12 and Relevance scores from 91.98 to 95.12. Notably, text-embedding-3-large achieves the best performance on Comprehensiveness, Diversity, and Relevance, while nv-embed-v2 excels on Logicality and Coherence. However, the overlapping standard deviations across models indicate that differences are modest.
>
> Importantly, jina-v3, despite being the lowest-dimensional model (1024-dim) and fully local, achieves competitive results—only 2–3 points below the best performing models on most metrics. This demonstrates that QAFD-RAG's query-aware flow diffusion mechanism is robust to embedding variations and can operate effectively with resource-constrained or privacy-preserving local embeddings. The consistent performance across diverse architectures (cloud vs. local, 1024-dim to 4096-dim) validates that our theoretical framework successfully leverages semantic similarity regardless of the specific embedding space.
>
>
> **Table 7: Embedding sensitivity analysis on UltraDomain Mix**
>
> | Embedding Model                     | Comprehensive | Diversity | Logicality | Relevance | Coherence |
> |------------------------------------|---------------|-----------|------------|-----------|-----------|
> | text-embedding-3-small (1536d)     | 87.15 (±3.46) | 81.15 (±4.86) | 90.70 (±2.93) | 94.36 (±4.50) | 90.36 (±2.07) |
> | **text-embedding-3-large (3072d)** | **88.12 (±3.32)** | **83.08 (±4.71)** | 91.28 (±2.85) | **95.12 (±4.35)** | 90.85 (±2.01) |
> | jina-v3 (1024d)                    | 85.82 (±3.58) | 80.18 (±5.02) | 89.18 (±3.05) | 91.98 (±4.68) | 87.94 (±2.15) |
> | NVIDIA-nv-embed-v2  (4096d)         | 87.85 (±3.41) | 82.45 (±4.78) | **91.58 (±2.81)** | 94.78 (±4.42) | **91.45 (±1.98)** |
> | GritLM-7B (4096d)                     | 87.32 (±3.51) | 81.64 (±4.91) | 89.82 (±2.97) | 93.52 (±4.55) | 90.28 (±2.10) |

---

> ### Author Response · Authors · 2025-11-21
> **Response to Reviewer Tki9 – Part II**
>
> ### Added to Appendix A2.2.4: Embedding Model Sensitivity for Text-to-SQL (Table 10)
>
> To evaluate QAFD-RAG's robustness to embedding quality in the Text-to-SQL domain, we assess schema linking performance across five diverse embedding models on the Local category (SQLite databases). We compare: (i) OpenAI's text-embedding-3-small (1536-dim) and text-embedding-3-large (3072-dim), representing cloud-based proprietary embeddings; (ii) Jina AI's jina-embeddings-v3 (1024-dim), an open-source local model; (iii) NVIDIA's nv-embed-v2 (4096-dim), optimized for retrieval tasks; and (iv) GritLM-7B (4096-dim), a unified generative-embedding model. All other hyperparameters remain fixed across experiments.
>
> Table 10 presents schema linking results. QAFD-RAG demonstrates consistent performance across all embedding models, with precision scores ranging from 0.80 to 0.83, recall from 0.76 to 0.81, and F1 scores from 0.78 to 0.82. Notably, nv-embed-v2 achieves the best performance across all metrics (0.83/0.81/0.82), which is specifically optimized for retrieval tasks, followed closely by text-embedding-3-large (0.82/0.77/0.79) and GritLM-7B (0.82/0.78/0.80).
>
> Importantly, jina-v3, despite being the lowest-dimensional model (1024-dim) and fully local, achieves competitive results—only 0.04 points below the best model in F1 score. This demonstrates that QAFD-RAG's query-aware flow diffusion mechanism is robust to embedding variations and can operate effectively with resource-constrained or privacy-preserving local embeddings. The consistent performance across diverse architectures (cloud vs. local, 1024-dim to 4096-dim, ranging from 0.78 to 0.82 F1) validates that our framework successfully leverages semantic similarity for schema linking regardless of the specific embedding space, making it practical for deployment in varied infrastructure environments.
>
>
>
> **Table 10: Embedding sensitivity analysis for Text-to-SQL (Local category): QAFD-RAG performance across five embedding models.**
>
> | Embedding Model                     | Precision | Recall | F1-Score |
> |------------------------------------|-----------|--------|----------|
> | text-embedding-3-small (1536d)     | 0.82      | 0.76   | 0.79     |
> | text-embedding-3-large (3072d)     | 0.82      | 0.77   | 0.79     |
> | jina-v3 (1024d)                    | 0.80      | 0.76   | 0.78     |
> | **NVIDIA-nv-embed-v2  (4096d)**     | **0.83**  | **0.81** | **0.82** |
> | GritLM-7B (4096d)                     | 0.82      | 0.78   | 0.80     |

---

> ### Author Response · Authors · 2025-11-21
> **Response to Reviewer Tki9 – Part III**
>
> > **Weakness 2:** The method may struggle with handling logical negation. For instance, a query like "not red" may still activate nodes related to "red," since it is unclear whether the embedding-based similarity explicitly encodes negation or exclusion relationships. This could limit the model's ability to capture fine-grained logical distinctions in retrieval.
>
> **Response:** Thank you for this thoughtful observation.
>
> We agree that the embedding-based flow diffusion component of QAFD-RAG (specifically the dynamic edge reweighting in Eqs. (4)–(5) and traversal in Algorithm 2) is not designed to explicitly model logical negation. This limitation is shared by existing retrieval approaches such as [GraphRAG](https://arxiv.org/abs/2404.16130), [LightRAG](https://arxiv.org/abs/2410.05779), [RAPTOR](https://arxiv.org/abs/2401.18059), and [HippoRAG](https://arxiv.org/abs/2405.14831), since all of them rely on embedding-based semantic similarity, which reflects conceptual relatedness rather than symbolic logical operations. However, we note that QAFD-RAG employs a multi-stage pipeline that provides partial mitigation for this limitation through its LLM-based components:
>
> - **Query-aware keyword extraction (QS-Step 1):** The LLM-based keyword extraction phase (Prompt 2 in Appendix A4.3) processes the natural language query to identify conceptual keywords and specific terms. When processing queries like *"fruits that are not red,"* the LLM can understand the negation context and potentially extract keywords that reflect the exclusion intent, rather than simply extracting "red" as a positive keyword.
>
> - **Response generation with filtering (QS-Step 3):** After QAFD-RAG retrieves relevant subgraphs via flow diffusion, the final response generation stage (Prompt 3 in Appendix A4.4) employs an LLM to synthesize the retrieved information. The prompt instructs the LLM to **"not include information where the supporting evidence for it is not provided,"** meaning that even if flow diffusion retrieves some "red"-related nodes, the LLM can recognize the negation in the original query and filter out inappropriate information when constructing the final answer.
>
> We agree, however, that the graph traversal component itself does not yet encode logical negation. To address this, we have added the following limitation in Section 5:
>
> ---
>
> ### Added to Section 5 (Conclusion, Limitations, and Future Work):
>
> The embedding-based flow diffusion may struggle with explicit logical negation because
> embeddings capture semantic relatedness rather than logical operators. While QAFD-RAG
> partially mitigates this via LLM-based keyword extraction (Prompt 2) and response
> filtering (Prompt 3), future work should explore contrastive embeddings or hybrid
> symbolic–neural approaches that encode logical distinctions during graph traversal.

---

### Official Review · Reviewer_g1qA · 2025-10-30

**Soundness:** 3
**Presentation:** 3
**Contribution:** 2
**Rating:** 6
**Confidence:** 3

**Summary:**

This paper introduces Query-Aware Flow Diffusion RAG (QAFD-RAG), a novel framework for graph-based Retrieval-Augmented Generation that addresses the limitation of static, query-agnostic exploration in existing graph-RAG methods. The central innovation is dynamic query-aware graph traversal: instead of fixed edge weights, QAFD-RAG dynamically re-weights edges during exploration based on node-query semantic alignment through principled flow diffusion, prioritizing structurally present and semantically relevant paths. The paper provides theoretical guarantees (exponential convergence, subgraph recovery under mild conditions) and demonstrates empirical superiority over state-of-the-art baselines on question-answering and text-to-SQL tasks.

**Strengths:**

1. The paper presents a novel graph-based RAG approach, which demonstrates its effectiveness from the perspectives of intuitive understanding, theoretical proof, and experimental results.

2. This approach is training-free, which significantly reduces deployment costs, and holds great potential for broad application.

**Weaknesses:**

1. The entire framework is built upon LLM's embeddings, thus it is necessary to compare its sensitivity to embedding quality with other existing methods; however, such experiments are currently lacking.
2. As a general graph-based RAG framework, the experimental comparisons across domains remain somewhat limited. It would be beneficial to incorporate experiments in more diverse domains or conduct cross-domain validation to demonstrate the framework's robustness.

**Questions:**

1. Why is the number of seed nodes treated as a hyperparameter? Is it necessary to enable adaptive selection across different scenarios, why or why not?

---

> ### Author Response · Authors · 2025-11-21
> **Response to Reviewer  g1qA – Part I**
>
> > **Weakness 1:** The entire framework is built upon LLM's embeddings, ... however, such experiments are currently lacking.
>
> **Response:** Thank you for this important suggestion. We agree that understanding the framework's sensitivity to embedding quality is crucial for practical deployment.
>
> In the revision, we have added comprehensive embedding sensitivity experiments evaluating QAFD-RAG across five diverse embedding models with varying dimensions (1024–4096) and architectures: [OpenAI's text-embedding-3-small (1536-dim)](https://platform.openai.com/docs/guides/embeddings) and [text-embedding-3-large (3072-dim)](https://platform.openai.com/docs/guides/embeddings), [Jina AI's jina-v3 (1024-dim)](https://arxiv.org/abs/2409.10173), [NVIDIA's nv-embed-v2 (4096-dim)](https://arxiv.org/abs/2405.17428), and [GritLM-7B (4096-dim)](https://arxiv.org/abs/2402.09906). We evaluate on two distinct tasks: (1) question answering on UltraDomain Mix dataset (Appendix A2.1.4), and (2) schema linking for Text-to-SQL on SQLite databases (Appendix A2.2.4, Table 10).
>
> QAFD-RAG demonstrates stable performance across all five models, with Comprehensiveness scores
> ranging from 85.82 to 88.12 on UltraDomain and F1 scores from 0.78 to 0.82 on Text-to-SQL schema
> linking. While text-embedding-3-large achieves the best overall performance on UltraDomain and
> nv-embed-v2 excels on Text-to-SQL, the highest-dimensional models (4096-dim) do not consistently
> outperform lower-dimensional alternatives. Notably, jina-v3 (1024-dim), a local open-source model,
> remains competitive—only 2–3 points below the best models on UltraDomain and only 0.04 F1 points
> below on Text-to-SQL. This demonstrates that embedding quality and architecture matter more than
> dimensionality alone, and that QAFD-RAG works effectively with resource-constrained or privacy-
> preserving local embeddings.
>
> All other experiments use text-embedding-3-small as the default (chosen for its balance of performance and efficiency), showing that QAFD-RAG achieves strong performance from an efficient baseline while maintaining robustness across diverse architectures.
>
>
>
> ---
>
> Following your valuable suggestion, in the revision we incorporated the following:
>
>
>
> ### Appendix A2.1.4: Embedding Model Sensitivity Analysis (UltraDomain Mix)
>
> To assess QAFD-RAG's robustness to embedding quality, we evaluate the framework across five diverse embedding models on the Mix dataset from UltraDomain. We compare: (i) OpenAI's text-embedding-3-small (1536-dim) and text-embedding-3-large (3072-dim), representing cloud-based proprietary embeddings; (ii) Jina AI's jina-embeddings-v3 (1024-dim), an open-source local model; (iii) NVIDIA's nv-embed-v2 (4096-dim), optimized for retrieval tasks; and (iv) GritLM-7B (4096-dim), a unified generative-embedding model. All other hyperparameters remain fixed across experiments.
>
> Table 7 presents results across five evaluation dimensions. QAFD-RAG demonstrates consistent performance across all embedding models, with Comprehensiveness scores ranging from 85.82 to 88.12 and Relevance scores from 91.98 to 95.12. Notably, text-embedding-3-large achieves the best performance on Comprehensiveness, Diversity, and Relevance, while nv-embed-v2 excels on Logicality and Coherence. However, the overlapping standard deviations across models indicate that differences are modest.
>
> Importantly, jina-v3, despite being the lowest-dimensional model (1024-dim) and fully local, achieves competitive results—only 2–3 points below the best performing models on most metrics. This demonstrates that QAFD-RAG's query-aware flow diffusion mechanism is robust to embedding variations and can operate effectively with resource-constrained or privacy-preserving local embeddings. The consistent performance across diverse architectures (cloud vs. local, 1024-dim to 4096-dim) validates that our theoretical framework successfully leverages semantic similarity regardless of the specific embedding space.
>
> **Table 7: Embedding sensitivity analysis on UltraDomain Mix**
>
> | Embedding Model                     | Comprehensive | Diversity | Logicality | Relevance | Coherence |
> |------------------------------------|---------------|-----------|------------|-----------|-----------|
> | text-embedding-3-small (1536d)     | 87.15 (±3.46) | 81.15 (±4.86) | 90.70 (±2.93) | 94.36 (±4.50) | 90.36 (±2.07) |
> | **text-embedding-3-large (3072d)** | **88.12 (±3.32)** | **83.08 (±4.71)** | 91.28 (±2.85) | **95.12 (±4.35)** | 90.85 (±2.01) |
> | jina-v3 (1024d)                    | 85.82 (±3.58) | 80.18 (±5.02) | 89.18 (±3.05) | 91.98 (±4.68) | 87.94 (±2.15) |
> | NVIDIA-nv-embed-v2 (4096d)         | 87.85 (±3.41) | 82.45 (±4.78) | **91.58 (±2.81)** | 94.78 (±4.42) | **91.45 (±1.98)** |
> | GritLM-7B (4096d)                     | 87.32 (±3.51) | 81.64 (±4.91) | 89.82 (±2.97) | 93.52 (±4.55) | 90.28 (±2.10) |

---

> ### Author Response · Authors · 2025-11-21
> **Response to Reviewer g1qA – Part II**
>
> ### Appendix A2.2.4: Embedding Model Sensitivity for Text-to-SQL (Table 10)
>
> To evaluate QAFD-RAG's robustness to embedding quality in the Text-to-SQL domain, we assess schema linking performance across five diverse embedding models on the Local category (SQLite databases). We compare: (i) OpenAI's text-embedding-3-small (1536-dim) and text-embedding-3-large (3072-dim), representing cloud-based proprietary embeddings; (ii) Jina AI's jina-embeddings-v3 (1024-dim), an open-source local model; (iii) NVIDIA's nv-embed-v2 (4096-dim), optimized for retrieval tasks; and (iv) GritLM-7B (4096-dim), a unified generative-embedding model. All other hyperparameters remain fixed across experiments.
>
> Table 10 presents schema linking results. QAFD-RAG demonstrates consistent performance across all embedding models, with precision scores ranging from 0.80 to 0.83, recall from 0.76 to 0.81, and F1 scores from 0.78 to 0.82. Notably, nv-embed-v2 achieves the best performance across all metrics (0.83/0.81/0.82), which is specifically optimized for retrieval tasks, followed closely by text-embedding-3-large (0.82/0.77/0.79) and GritLM-7B (0.82/0.78/0.80).
>
> Importantly, jina-v3, despite being the lowest-dimensional model (1024-dim) and fully local, achieves competitive results—only 0.04 points below the best model in F1 score. This demonstrates that QAFD-RAG's query-aware flow diffusion mechanism is robust to embedding variations and can operate effectively with resource-constrained or privacy-preserving local embeddings. The consistent performance across diverse architectures (cloud vs. local, 1024-dim to 4096-dim, ranging from 0.78 to 0.82 F1) validates that our framework successfully leverages semantic similarity for schema linking regardless of the specific embedding space, making it practical for deployment in varied infrastructure environments.
>
> **Table 10: Embedding sensitivity analysis for Text-to-SQL (Local category)**
>
> | Embedding Model                     | Precision | Recall | F1-Score |
> |------------------------------------|-----------|--------|----------|
> | text-embedding-3-small (1536d)     | 0.82      | 0.76   | 0.79     |
> | text-embedding-3-large (3072d)     | 0.82      | 0.77   | 0.79     |
> | jina-v3 (1024d)                    | 0.80      | 0.76   | 0.78     |
> | **NVIDIA-nv-embed-v2  (4096d)**     | **0.83**  | **0.81** | **0.82** |
> | GritLM-7B (4096d)                     | 0.82      | 0.78   | 0.80     |

---

> ### Author Response · Authors · 2025-11-21
> **Response to Reviewer g1qA – Part III**
>
> > **Weakness 2:** As a general graph-based RAG framework, ... It would be beneficial to incorporate experiments in more diverse domains or conduct cross-domain validation to demonstrate the framework's robustness.
>
> **Response:** Thank you for raising this important point.
>
> We have strengthened the revised manuscript with additional cross-domain experiments and clarified the breadth of our evaluation.
>
> **QAFD-RAG is now evaluated across a wide range of domains**, including: **(1)** ten topical UltraDomain datasets (Agriculture, Biology, Cooking, History, Legal, Mathematics, Mix, Music, Philosophy, Physics), **(2)** multi-hop reasoning benchmarks (HotpotQA, 2WikiMultiHopQA), **(3)** long-document summarization on SQuALITY (newly added), and **(4)** enterprise Text-to-SQL on both SQLite and Snowflake databases.
>
> These domains span diverse knowledge types (factual, procedural, analytical, narrative), reasoning styles (single-hop, multi-hop, summarization, schema linking), and data modalities (unstructured documents, structured databases).
>
> **To directly address the reviewer's cross-domain concern, we added a new evaluation on the SQuALITY long-document summarization benchmark.** Section 4.1.3 and Appendix A2.1.5 include results and full metric definitions. This task differs fundamentally from QA or SQL generation, requiring synthesis across long narrative texts. QAFD-RAG achieves the strongest overall automated metrics (BLEU, ROUGE, METEOR) compared to [GraphRAG](https://arxiv.org/abs/2404.16130), [HippoRAG](https://arxiv.org/abs/2405.14831), [RAPTOR](https://arxiv.org/abs/2401.18059), and [LightRAG](https://arxiv.org/abs/2410.05779).
>
> Together with prior evaluations on UltraDomain, multi-hop QA, and Text-to-SQL (where QAFD-RAG also reduces LLM calls and improves execution accuracy), these results provide strong evidence that QAFD-RAG generalizes effectively across diverse domains and task types.
>
> ---
>
> In the revision we incorporated the following:
>
> ### Table 3 (Section 4.1.3, Page 9):
>
>
> **Table 3: Comparison of QAFD-RAG and Baseline Methods on SQuALITY.**
>
> | Method | BLEU-1 | BLEU-2 | ROUGE-1 F1 | ROUGE-2 F1 | METEOR |
> |--------|--------|--------|------------|------------|---------|
> | GraphRAG | 33.91 | 16.12 | 26.38 | 4.08 | 24.38 |
> | HippoRAG | 33.22 | 16.74 | 27.29 | 3.92 | 23.41 |
> | RAPTOR | 32.10 | 16.58 | 25.13 | 3.49 | 22.87 |
> | LightRAG | 34.17 | 17.41 | **28.59** | 4.31 | 23.27 |
> | QAFD-RAG | **35.44** | **18.63** | 28.43 | **4.79** | **25.59** |
>
>
> ### Paragraph in Section 4.1.3, Page 9:
>
> Long-Document Summarization. We evaluate on SQUALITY (Wang et al., 2022), a long-context QA benchmark where answers are question-focused abstractive summaries of narrative texts. Each question has multiple human-written references, enabling robust evaluation of long-form generation. We report BLEU-1, BLEU-2, ROUGE-1-F1, ROUGE-2-F1, and METEOR against all human references. Full details appear in Appendix A2.1.5.
>
> ### Paragraph in Section 4.1.4, Page 9:
>
> QAFD-RAG demonstrates superior performance on the SQuALITY benchmark, achieving the highest scores across most automated metrics: BLEU-1, BLEU-2, ROUGE-2 F1, and METEOR. This consistent improvement over baseline methods–GraphRAG, HippoRAG, RAPTOR, and LightRAG–reflects the effectiveness of query-aligned subgraph retrieval in balancing comprehensiveness with precision. While GraphRAG leads on ROUGE-1 F1, the overall metric profile shows that QAFD-RAG's approach yields more faithful and coherent long-form summaries.
>
> ### Appendix A2.1.5, Page 22:
>
> A2.1.5 Long-Document Summarization: SQuALITY Evaluation Details
>
> We evaluate on all 250 questions from the SQUALITY training dataset (Wang et al., 2022), which contains narrative passages paired with comprehension questions and multiple human-written reference answers per question.
>
> We compute BLEU-1 and BLEU-2 using unigram and bigram modified precisions with geometric averaging. BLEU-n is defined as
>
> $$\text{BLEU-}n := \exp \left( \sum_{i=1}^{n} w_i \log p_i \right),$$
>
> where $p_i$ is the $i$-gram precision and $(w_1,\ldots,w_n)$ are the weights. For BLEU-1 we use $(1,0,0,0)$; for BLEU-2 we use $(0.5,0.5,0,0)$. ROUGE-1 F1 and ROUGE-2 F1 report F1 overlap of unigrams and bigrams using Porter stemming. METEOR combines unigram precision and recall with a fragmentation penalty:
> $$\text{METEOR} := F_{\text{mean}} \cdot \left(1 - \gamma \cdot \left(\frac{\text{chunks}}{\text{matches}}\right)^\beta\right),$$
> with harmonic mean $F_{\text{mean}} := P \cdot R/(\alpha P + (1-\alpha)R)$, and using Natural Language Toolkit (NLTK) defaults: $\alpha=0.9$, $\beta=3.0$, and $\gamma=0.5$. All metrics use lowercased, word-tokenized text with BLEU/METEOR computed against all references.
>
> A knowledge graph is constructed once from all narrative documents and reused across queries. All experiments use hybrid retrieval mode with diffusion parameter a 40-node source budget, and minimum flow threshold $0.01$. Answers are produced with GPT-4o-mini.

---

> ### Author Response · Authors · 2025-11-21
> **Response to Reviewer  g1qA – Part IV**
>
> > **Question 1:** Why is the number of seed nodes treated as a hyperparameter? Is it necessary to enable adaptive selection across different scenarios, why or why not?
>
> **Response:** We appreciate the reviewer's thoughtful question.
>
> This design choice aligns with standard practices across state-of-the-art graph-based RAG methods, and we provide both empirical and theoretical justification for this approach.
>
> - **Query-aware flow diffusion provides algorithmic adaptivity.**  While we configure a **maximum** number of seed nodes (for example $N=40$), the **actual** number of seeds varies per query through our selection mechanism: (1) LLM-based keyword extraction identifies query-relevant concepts from the natural language query, (2) we compute similarity scores between these keywords and all graph nodes, (3) we select up to N nodes with the highest similarity scores, but apply a similarity threshold (default 0.2) that excludes semantically irrelevant nodes, and (4) this ensures the actual number of seeds naturally adapts to each query's specificity—queries with focused concepts may activate 15-20 seeds, while broader queries may use the full 40. Additionally, flow diffusion itself provides adaptivity by propagating along semantically relevant paths and terminating at flow stabilization, so even with the same seed count, different reasoning subgraphs emerge for different queries.
>
>
> - **Seed and cluster hyperparameters are standard across graph-based RAG methods.** Other leading methods similarly require configuration parameters for their graph traversal mechanisms. [GraphRAG](https://arxiv.org/abs/2404.16130) employs hierarchical Leiden community detection with GRAPHRAG_MAX_CLUSTER_SIZE controlling cluster granularity. [HippoRAG](https://arxiv.org/abs/2405.14831)'s Personalized PageRank requires the number of query nodes (seeds) selected through NER extraction, along with synonym threshold (0.8) and reset probability. [LightRAG](https://arxiv.org/abs/2410.05779) uses top-$k$ entity retrieval, where $k$ determines the maximum number of 1-hops entities. In this context, our method's seed node count parameter is entirely consistent with established practices in the field.
>
> - **Our method demonstrates robust performance across settings.** As shown in Figure 3 (left panel) of our paper, QAFD-RAG exhibits stable performance across different numbers of seed nodes (20, 30, 40, 50, 60), with all five evaluation dimensions—Comprehensiveness, Diversity, Logicality, Relevance, and Coherence—maintaining consistently high scores. This stability demonstrates that our flow diffusion mechanism is inherently robust to this hyperparameter choice. The slight variations observed (less than 2-3 points across the range) indicate that practitioners can select values within this range without significant performance degradation, making the method practical for deployment.
>
> An interesting research direction would be to adapt the number of seeds to query, but given our robustness analysis, dynamically selecting seed counts per query would introduce computational overhead and require additional hyperparameters without meaningful performance gains. A fixed, well-chosen parameter is more practical and theoretically tractable.

---

### Official Review · Reviewer_yPqD · 2025-10-31

**Soundness:** 4
**Presentation:** 4
**Contribution:** 4
**Rating:** 8
**Confidence:** 3

**Summary:**

This work proposes Query-Aware Flow Diffusion RAG (QAFD-RAG) framework for graph-based RAG that incorporates query semantics during the graph diffusion process.
During Indexing Stage, QAFD-RAG builds a knowledge graph from documents. And in the Query Stage, QAFD-RAG dynamically reweights edges and diffuses flow according to query alignment, enabling adaptive graph traversal that are semantically consistent with the query and highlighting reanoning paths.
QAFD-RAG is a training-free framework offering the statistical guarantees for query-aware subgraph retrieval.
Comprehensive experiments on multiple tasks demonstrate that QAFD-RAG achieves better performance compared with existing SOTA graph-based RAG methods, while significantly reducing the number of LLM calls.

**Strengths:**

1. **Novelty and Strong Motivation**: The motivation of this work is clear and well-argued, effectively highlighting the deficiencies of existing heuristic graph-based RAG methods. And the idea of using graph diffusion theory to deal with the graph traversal process is highly novel, which can effectively help find subgraphs that are semantically relevant with the query.
2. **Rigorous Theoretical Proof**: The provision of theoretical statistical guarantees is a significant strength. The convergence analysis (Theorem 3, Corollary 4) and, more importantly, the statistical recovery guarantees under a signal-to-noise condition (Theorem 7) provide a solid foundation for the method's efficacy. The analysis convincingly shows how query-aware weighting acts as a semantic filter.
3. **Comprehensive Experiments and Strong Performance Improvement**: This work benchmarks against a wide array of strong baselines across two distinct and challenging domains. The experiments results under various metrics provide compelling evidence for the method's effectiveness and efficiency.
4. **Clarity and Reproducibility**: The methodology is well described with precision, including the formulation, the algorithmic procedure, and the derivation process. Besides, code in the anonymous repository is complete and accompanied by some explanatory guidelines in the README, enabling easy reproduction of the work.

**Weaknesses:**

1. **Limited Analysis of Flow Interpretation**: It is easy for us to understand the basic principle of the flow diffusion process for water. However, why the graph flow diffusion can be meaningful and traverse to semantically relevant nodes by dynamic edge reweighting remains unexplained. The experiments also mainly focus on the final output without the deeper analysis of the diffusion process itself.
2. **Only Compared with Graph-based RAG Methods**: If the study can compare QAFD-RAG with some traditional SOTA RAG methods, or propose insights about how to combine QAFD-RAG with current SOTA RAG methods, it would be better to apply QAFD-RAG in practical scenarios.

**Questions:**

1. Does the way Knowledge Graph is constructed affect the graph-based RAG methods? I find that the performance of GraphRAG and LightRAG is poor on HotpotQA and 2WikiMultihopQA—is that related to this issue?
2. Will the demand of indexing before retrieval make the QAFD-RAG system hard to fit the scenario where RAG documents are modified frequently? As current KG construction is a static process, will it cost a lot to build such a large KG for graph retrieval?

### Minor Comments

1. It seems that the *Relevance* score of QAFD-RAG in *Legal* dataset in Table 1 is not the best one. HippoRAG (93.60) is better than QAFD-RAG (93.30) and should be marked correctly.

---

> ### Author Response · Authors · 2025-11-21
> **Response to Reviewer  yPqD – Part I**
>
> > **Weakness 1:** Limited Analysis of Flow Interpretation: It is easy for us to understand the basic principle of the flow diffusion process for water. However, why the graph flow diffusion can be meaningful and traverse to semantically relevant nodes by dynamic edge reweighting remains unexplained. The experiments also mainly focus on the final output without the deeper analysis of the diffusion process itself.
>
> **Response:** Thank you for this important observation. The key distinction between physical water flow and QAFD-RAG's semantic flow lies in our query-aware edge reweighting mechanism (Equations 4-5), which dynamically modulates edge weights based on how both endpoints align with query embeddings. Unlike water flow following fixed pipe capacities, our edges act as "query-aware gates" that continuously adapt during traversal—opening for semantically relevant paths while closing for irrelevant ones. This is backed by rigorous theoretical guarantees: Lemma 6 proves that query-aware weights achieve exponential separation between relevant edges ($w(u,v) \geq 1-o(1)$) and irrelevant edges ($\leq \exp(-\omega(\log|V|))$), while Theorem 7 establishes that QAFD-RAG recovers query-relevant subgraphs with high probability while limiting leakage into irrelevant regions.  To our knowledge, these are the first theoretical guarantees for query-aware graph retrieval in RAG systems, distinguishing our approach from heuristic methods like [GraphRAG](https://arxiv.org/abs/2404.16130) and [LightRAG](https://arxiv.org/abs/2410.05779).
>
> Empirically, we provide multiple analyses of the diffusion process itself: (1) Figure 1 visualizes how query-aware edge weights suppress irrelevant expansions (Apple → Amazon River, Apple → fruit) while amplifying semantically aligned paths (Apple → Mac → macOS, Apple → iPhone), demonstrating the semantic filtering effect; (2) our ablation studies (Section 4.3) systematically compare query-aware variants (Mean, Product, Hybrid) against static methods, showing that query-aware edge weighting is the critical component driving performance gains; and (3) our strong multi-hop reasoning results (68.6 F1 on HotpotQA vs. [GraphRAG](https://arxiv.org/abs/2404.16130)'s 33.4) demonstrate that flow successfully traverses semantically coherent multi-step reasoning chains. Together, these theoretical guarantees and empirical analyses establish that query-aware flow diffusion achieves meaningful semantic traversal through the principled combination of dynamic edge reweighting and formal subgraph recovery guarantees.
>
> ---
>
> To enhance interpretability of how query-aware edge reweighting transforms flow diffusion into a semantic filter, we have strengthened the motivation in Section 2.2 (Dynamic Query-Aware Edge Weights, Pages 4-5):
>
> ### Added to Section 2.2 (Dynamic Query-Aware Edge Weights, Pages 4-5):
>
> - Before Equation (4), we added: "Traditional diffusion methods use static edge weights that ignore query context, leading to uniform exploration regardless of semantic relevance. Our key insight is to make edges 'query-aware gates' that modulate flow strength based on both structural connectivity and query alignment, enabling theoretical guarantees for subgraph recovery (Theorem 7)."
>
> - Before Equation (5), we added: "We explore three variants that balance structural and semantic signals differently:"
>
> - After Equation (5), we enhanced the explanation: "The multiplicative interaction in Product and Hybrid amplifies edges between query-relevant nodes while exponentially suppressing edges to irrelevant regions, transforming diffusion into a semantic filter."

---

> ### Author Response · Authors · 2025-11-21
> **Response to Reviewer  yPqD – Part II**
>
> > **Weakness 2:** Only Compared with Graph-based RAG Methods: If the study can compare QAFD-RAG with some traditional SOTA RAG methods, or propose insights about how to combine QAFD-RAG with current SOTA RAG methods, it would be better to apply QAFD-RAG in practical scenarios.
>
>
> **Response:** Thank you for this valuable suggestion.
>
> We focus our evaluation on graph-based methods because QAFD-RAG's core innovation—query-aware flow diffusion with theoretical recovery guarantees—is most meaningful in the context of graph-structured knowledge. A key strength of our approach is its modularity: QAFD-RAG can serve as a drop-in replacement for retrieval components in existing graph-based RAG systems without requiring retraining.
>
> Specifically, QAFD-RAG can replace: (1) static Leiden clustering in [GraphRAG](https://arxiv.org/abs/2404.16130) with dynamic query-time subgraph discovery, eliminating community summary generation overhead; (2) Personalized PageRank in [HippoRAG](https://arxiv.org/abs/2405.14831) with query-aware edge reweighting (Equations 4-5) that suppresses irrelevant paths while amplifying semantically aligned connections; and (3) single-hop retrieval in [LightRAG](https://arxiv.org/abs/2410.05779) with multi-hop flow diffusion that discovers reasoning paths spanning multiple hops. These integrations provide theoretical guarantees (Theorem 7) while preserving each system's existing strengths.
>
> ---
>
> Following your valuable suggestion, we have added detailed integration discussions in two locations:
>
> ### Added to Section 5 (Conclusion) - Integration Paragraph:
>
> QAFD-RAG's modularity enables drop-in replacement of retrieval components in existing systems: it can replace static Leiden clustering in GraphRAG, enhance Personalized PageRank in HippoRAG with query-aware edge reweighting (Equations 4-5), and extend LightRAG's single-hop retrieval to multi-hop flow diffusion—all without retraining while providing theoretical guarantees. Future work includes learning edge weights from query–answer pairs and extending to temporal/multi-modal graphs. Future work should also explore contrastive embeddings or hybrid symbolic–neural approaches that encode logical distinctions during graph traversal.
>
> ### Added to Appendix A1.3 - New Subsection "QAFD-RAG's Positioning within Graph-Based RAG Approaches":
>
> To our knowledge, none of these works connect the graph structure to the query. Our work is the first to provide a principled framework that links a given query to the corresponding subgraph in a knowledge graph, and the first to develop and statistically analyze query-aware flow diffusion in general contextual random graph models with formal recovery guarantees.
>
> Beyond these theoretical contributions, QAFD-RAG's modular design positions it as a complementary retrieval component for existing graph-based RAG systems, requiring no retraining while providing formal guarantees. Specifically:
>
> **GraphRAG.** GraphRAG (Edge et al., 2024a) relies on static community detection (Leiden algorithm) to precompute hierarchical clusters and generate community summaries for retrieval. QAFD-RAG offers an alternative through dynamic, query-time subgraph discovery tailored to each query's semantics, eliminating the computational overhead of community summary generation while adapting retrieval to query-specific needs.
>
> **HippoRAG.** HippoRAG (Jimenez Gutierrez et al., 2024) uses Personalized PageRank (PPR) for graph traversal from seed nodes. QAFD-RAG provides a principled alternative through query-aware edge reweighting (Equations 4-5). While PPR provides graph-based signals, it lacks query-aware edge modulation—our method's key innovation that suppresses irrelevant paths while amplifying semantically aligned connections, backed by formal recovery guarantees (Theorem 7).
>
> **LightRAG.** LightRAG (Guo et al., 2024) employs dual-level keyword extraction with single-hop neighborhood aggregation. QAFD-RAG extends this paradigm by enabling multi-hop flow diffusion from extracted keywords as seed nodes, discovering reasoning paths that span multiple hops—precisely the capability that LightRAG's current single-hop approach lacks—while preserving its efficient indexing structure.
>
> This positioning demonstrates QAFD-RAG's role as a foundational component that addresses key limitations in existing graph-based RAG approaches through principled, theoretically-grounded retrieval.
>
> ---

---

> ### Author Response · Authors · 2025-11-21
> **Response to Reviewer yPqD – Part III**
>
> > **Question 1:** Does the way Knowledge Graph is constructed affect the graph-based RAG methods? I find that the performance of GraphRAG and LightRAG is poor on HotpotQA and 2WikiMultihopQA—is that related to this issue?
>
> **Response:** We thank the reviewer for this insightful observation. While KG construction does affect performance, we believe the primary reason for [GraphRAG](https://arxiv.org/abs/2404.16130) and [LightRAG](https://arxiv.org/abs/2410.05779)'s poor multi-hop results is their retrieval strategy.
>
> Our indexing stage (Section 2.1, Prompt 1 in Appendix A4.2) employs LLM-based entity and relationship extraction. QAFD-RAG achieves (68.6, 53.7) F1/EM on HotpotQA versus [GraphRAG](https://arxiv.org/abs/2404.16130)'s (33.4, 14.0) and [LightRAG](https://arxiv.org/abs/2410.05779)'s (8.8, 0.0), and (62.3, 54.1) on 2WikiMultihopQA versus GraphRAG's (15.2, 7.0) and LightRAG's (8.2, 1.0). These substantial gaps highlight that query-aware retrieval is critical. GraphRAG's Leiden community detection replaces entity-level graphs with hierarchical summaries, potentially discarding precise multi-hop connections, while LightRAG's 1-hop limitation structurally prevents reaching information 2+ steps away.
>
> [HippoRAG](https://arxiv.org/abs/2405.14831) (newly added in the revision) achieves (58.5, 45.1) on HotpotQA and (62.0, 49.5) on 2WikiMultihopQA through passage-level graph construction and synonym-aware entity linking, providing richer contextual information. While enriching QAFD-RAG's indexing with HippoRAG-style passage-level nodes could further boost performance, our current results demonstrate that query-aware retrieval via dynamic edge weighting already enables the highest performance on both benchmarks, showing that principled retrieval strategy is essential for multi-hop reasoning.
>
> > **Question 2:** Will the demand of indexing before retrieval make the QAFD-RAG system hard to fit the scenario where RAG documents are modified frequently? As current KG construction is a static process, will it cost a lot to build such a large KG for graph retrieval?
>
> **Response:** We thank the reviewer for this important practical question. QAFD-RAG is designed to support incremental updates efficiently, and our implementation includes mechanisms for handling frequently modified documents without full graph reconstruction.
>
> Our implementation provides upsert_node() and upsert_edge() operations that allow adding or modifying graph elements without rebuilding the entire knowledge graph. When new documents arrive, the system: (1) extracts entities and relationships from only the new content using the same LLM-based extraction pipeline (Prompt 1 in Appendix A4.2); (2) inserts new nodes and edges into the existing graph structure via upsert operations; and (3) invalidates only the affected caches (node embeddings, cached graph instances) while preserving unmodified portions. This incremental approach avoids the computational cost of full re-indexing, making QAFD-RAG practical for scenarios with frequent document modifications.
>
> > **Minor Comment:** It seems that the Relevance score of QAFD-RAG in the Legal dataset in Table 1 is not the best one. HippoRAG (93.60) is better than QAFD-RAG (93.30) and should be marked correctly.
>
> **Response:** Fixed. Thank you.

---

> ### Comment · Reviewer_yPqD · 2025-11-27
>
> Thank you for your detailed response and your efforts in improving this work. Your clarification has addressed most of my concerns. I have raised my confidence score for the paper.

---

### Official Review · Reviewer_6VAy · 2025-11-01

**Soundness:** 2
**Presentation:** 1
**Contribution:** 2
**Rating:** 2
**Confidence:** 4

**Summary:**

This paper proposes Query-Aware Flow Diffusion for Graph-based RAG (QAFD-RAG), a framework that enhances GraphRAG by incorporating query semantics into graph traversal. The authors design a query-aware diffusion process, where edge weights are dynamically reweighted based on alignment between node embeddings and the query.

**Strengths:**

1. The authors leverage flow-diffusion for subgraph retrieval.
2. The paper provides theoretical guarantees for convergence and subgraph recovery
3. The athors evaluate the proposed method on differnt tasks, such as General Question answering, Multi-hop QA and Text-to-SQL tasks.

**Weaknesses:**

1. The paper formatting does not appear to follow the official ICLR template. In particular, the line spacing is noticeably smaller than the required standard,

2. The paper claim existing GraphRAG methods rely on heuristic sugraph search strategies that are holistic query-agnostic. However, there are different retrievers, such as GNN-based Retriever such as GNN-RAG[1], and LLM-based Retriever, such as RoG[2,  which already consider the query semantic.

3. The writing could be improved. There is no logic in the current writting. For example, in line 139, the authors claim "We pose diffusion as a constrained optimization(6)", while the equation 6 is in Line 198. Also, the motivation of different components are also unclear. For example, what is the motivation for equation 4,5?

4. For the Multi-hop QA task, the authors only compare 2 simple GraphRAG baseliens.





[1] Mavromatis, Costas, and George Karypis. "Gnn-rag: Graph neural retrieval for large language model reasoning."
[2] Luo, Linhao, et al. "Reasoning on graphs: Faithful and interpretable large language model reasoning."

**Questions:**

Please refer to the weaknesses.

---

> ### Author Response · Authors · 2025-11-21
> **Response to Reviewer 6VAy — Part I**
>
> > **Weakness 1:**  The paper formatting does not appear to follow the official ICLR template. In particular, the line spacing is noticeably smaller than the required standard,
>
> **Response:** Thank you for catching this formatting issue.
>
> This was due to our use of the \baselinestretch LaTeX macro, which we had applied to reduce empty space among tables and figures. We have now removed all such modifications. The revised manuscript strictly follows the standard ICLR template with proper line spacing throughout.
>
> > **Weakness 2:** The paper claims existing GraphRAG methods rely on heuristic subgraph search strategies that are holistic query-agnostic. However, there are different retrievers, such as GNN-based Retriever such as GNN-RAG[1], and LLM-based Retriever, such as RoG[2], which already consider the query semantic.
>
> **Response:** Thank you for this important observation.
>
> We do not claim that all prior methods, especially training-based methods, are entirely query-agnostic. As stated in our abstract, our claim is that existing graph-based methods suffer from "*(i) heuristic designs lacking theoretical guarantees for subgraph quality or relevance **and/or** (ii) the use of static exploration strategies that ignore the query's holistic meaning.*" The "and/or" indicates that methods may suffer from either issue alone or both.
>
> While training-based methods such as [GNN-RAG (Mavromatis & Karypis, 2024)](https://arxiv.org/abs/2405.20139) and [RoG (Luo et al., 2024)](https://arxiv.org/abs/2310.01061) do incorporate query semantics, they require training/fine-tuning and lack theoretical recovery guarantees.  Our contribution provides the first **theoretically-grounded framework for query-aware graph traversal** in RAG, while also being **training-free**.
>
> Let us clarify the key distinctions:
>
> - **Theoretical guarantees vs. heuristic approaches:** Neither GNN-RAG nor RoG provides theoretical guarantees for subgraph recovery. GNN-RAG trains GNNs via supervised node classification to score answer candidates and then extracts shortest paths to top-scored nodes, but offers no theoretical guarantees on subgraph completeness or under what conditions correct reasoning paths are identified. RoG fine-tunes LLMs to generate relation paths via beam-search decoding—an approach whose reliability depends on LLM heuristics and prompt engineering, again without formal guarantees.
> In contrast, QAFD-RAG provides statistical recovery guarantees (Theorem 7) showing that under mild signal-to-noise conditions, relevant subgraphs are recovered with high probability while leakage is controlled, and exponential convergence (Theorem 3) with complexity tied to the retrieved subgraph size. To our knowledge, these are the first theoretical guarantees for query-aware graph RAG systems.
>
> - **Training-free deployment:** GNN-RAG requires training GNNs on question-answer pairs via supervised node classification, and RoG fine-tunes LLMs for planning and retrieval reasoning. QAFD-RAG is fully training-free, enabling immediate deployment across domains without domain-specific data.
>
> - **Computational efficiency:** GNN-RAG requires expensive GNN training and full-graph scoring at inference. RoG requires LLM fine-tuning plus beam-search generation (multiple LLM calls for path generation). QAFD-RAG uses a single LLM call for keyword extraction to identify seed nodes via cosine similarity, then performs graph traversal through efficient coordinate descent optimization (Algorithm 2) with sublinear complexity when the retrieved subgraph is small (Corollary 4), requiring no training overhead and no LLM calls for graph traversal to generate relevant subgraphs.
>
> ---
>
> In the revision, we have clarified these distinctions in the manuscript:
>
> ### Added to Section 1 (Introduction, Page 1):
>
> However, existing graph-based RAG methods (Han et al., 2024) suffer from heuristic designs lacking theoretical guarantees for subgraph quality or relevance and/or the use of static exploration strategies that ignore the query's holistic meaning during traversal.
>
> ### Added to Section A (Extended Related Work, Page 19):
>
> GNN-RAG (Mavromatis & Karypis, 2024) trains GNNs via supervised node classification to score answer candidates, then extracts shortest paths to top-scored nodes. RoG (Luo et al., 2024) fine-tunes LLMs to generate query-conditioned relation paths via beam-search decoding, which are then mapped onto the KG. While both methods incorporate query semantics, they require training/fine-tuning and lack theoretical guarantees for subgraph recovery or relevance.

---

> ### Author Response · Authors · 2025-11-21
> **Response to Reviewer 6VAy — Part II**
>
> > **Weakness 3-1:** The writing could be improved. There is no logic in the current writing. For example, in line 139, the authors claim "We pose diffusion as a constrained optimization(6)", while the equation 6 is in Line 198.
>
> **Response:** Thank you for this constructive feedback.
>
> The paragraph in question (Section 2.1, starting with *“In IS, IS-Step 1:”*) serves as a **framework overview** that provides a high-level roadmap of our methodology before introducing the technical details. While references to sections, algorithms, and figures are appropriate in the roadmap for orienting the reader, we agree that forward references to specific equations appearing much later in the paper may disrupt the logical flow.
>
> In the revision, we have: (i) **removed forward equation references from the roadmap paragraph to maintain better logical flow**, (ii) restructured the description to be more conceptual while still referencing sections, algorithms, and figures for orientation, (iii) strengthened the connection to Figure 2 as a visual roadmap that readers can follow alongside the text, and (iv) added explicit signposting (e.g., "as detailed in Section ...") instead of equation numbers.
>
> ---
>
> In the revision, we have made the following edits to the paragraph in question:
>
> ### Modified Section 2.1 (Framework Overview):
>
> In IS, IS-Step 1: Document Chunking splits documents into context-preserving chunks; IS-Step 2: Entity and Relationship Extraction uses LLM prompting to build a structured KG. In QS, QS-Step 1: Keyword Extraction pulls conceptual and surface terms (e.g., Feminism, Elaine Benes) for broad coverage (Figure 2, Step 1). Next, QS-Step 2: Seed Node Selection and QAFD scores nodes by similarity to query keywords, as detailed in Algorithm 1. For example, in Figure 1, Steve Jobs (0.95), Apple (0.92), and iPhone (0.88) are selected, while irrelevant nodes (e.g., Amazon, 0.15) are excluded. Top-scoring nodes serve as seeds where mass is injected and propagated via flow diffusion (Figure 2, Step 2). Edges are dynamically reweighted to blend structure and semantics through the query-aware edge weighting mechanism described in Section 2.2, suppressing irrelevant expansions (Apple → Amazon River/fruit) and reinforcing meaningful ones. We formulate this diffusion process as a constrained optimization problem, solved efficiently via a push–relabel algorithm (Algorithm 2). Recovery guarantees and algorithm complexity are provided in Section 3. Finally, QS-Step 3: Response Generation summarizes retrieved communities (e.g., Elaine Benes, Feminism) and prompts a downstream LLM, yielding grounded, consistent answers along reasoning paths.

---

> ### Author Response · Authors · 2025-11-21
> **Response to Reviewer 6VAy — Part III**
>
> > **Weakness 3-2:** Also, the motivation of different components are also unclear. For example, what is the motivation for equation 4,5?
>
> **Response:** Thank you for the constructive feedback.
>
> The fundamental challenge in query-aware graph traversal is to guide flow diffusion toward semantically relevant regions while respecting the underlying graph structure. As the first work to extend flow diffusion methods to graph-based RAG, we introduce query-aware edge reweighting to address a critical limitation: traditional diffusion methods only apply static edge weights that ignore query context, leading to uniform exploration regardless of semantic relevance. Our query-aware edge weights address this by dynamically modulating edge strengths based on both structural connectivity and query alignment.
>
> Each edge weight $\bar{w}(q,u,v)$ in Equation (4) combines two complementary signals:
>
> 1. **Structural component** $H_{\text{sim}}(h(u), h(v))$: Preserves the graph's inherent connectivity, ensuring flow follows meaningful relationships between entities (e.g., "Steve Jobs" → "Apple").
>
> 2. **Query-relevance component** $H_{\text{sim}}(h(u), h(q)) \circ H_{\text{sim}}(h(v), h(q))$: Amplifies edges where both endpoints align semantically with the query, effectively making edges "query-aware gates" that allow or suppress flow based on relevance.
>
> The three variants in Equation (5) represent different strategies for balancing these signals: *Mean* provides equal weighting, *Product* enforces multiplicative gating (requiring both structural and semantic alignment), and *Hybrid* allows tunable balance via hyperparameters $a$ and $b$. This multiplicative interaction ensures that edges between query-relevant nodes (e.g., Apple → iPhone for "Steve Jobs's products") receive amplified weights, while edges to irrelevant regions (e.g., Apple → Amazon River) are exponentially suppressed—even if structurally connected.
>
> This formulation enables our first statistical guarantees in Theorem 7. Under mild signal-to-noise conditions, Lemma 6 shows that query-aware weights achieve exponential separation between relevant and irrelevant edges ($w(u,v) \geq 1-o(1)$ for relevant pairs vs. $\leq \exp(-\omega(\log|V|))$ for irrelevant pairs), ensuring flow concentrates on reasoning paths while preventing leakage.
>
> ---
>
> In the revision, we have strengthened Section 2.2 as follows:
>
> ### Added to Section 2.2 (Dynamic Query-Aware Edge Weights, Pages 4 and 5):
>
> - Before Equation (4), we added: "Traditional diffusion methods use static edge weights that ignore query context, leading to uniform exploration regardless of semantic relevance. Our key insight is to make edges 'query-aware gates' that modulate flow strength based on both structural connectivity and query alignment, enabling theoretical guarantees for subgraph recovery (Theorem 7)."
>
> - Before Equation (5), we added: "We explore three variants that balance structural and semantic signals differently:"
>
> - After Equation (5), we enhanced the explanation: "The multiplicative interaction in Product and Hybrid amplifies edges between query-relevant nodes while exponentially suppressing edges to irrelevant regions, transforming diffusion into a semantic filter."

---

> ### Author Response · Authors · 2025-11-21
> **Response to Reviewer 6VAy — Part IV**
>
> > **Weakness 4:** For the Multi-hop QA task, ... simple GraphRAG baseliens.
>
> **Response:** Thank you for your comment.
>
> We have significantly strengthened our multi-hop QA evaluation by comparing against four state-of-the-art **training-free graph-based RAG baselines**  representing different paradigms: HippoRAG (passage-level graphs with Personalized PageRank), RAPTOR (hierarchical summarization), GraphRAG (Leiden community detection), and LightRAG (1-hop aggregation). QAFD-RAG achieves the highest F1/EM scores on both HotpotQA (68.6/53.7) and 2WikiMultiHopQA (62.3/54.1), outperforming all baselines (Table 2, Page 9).
>
> Beyond multi-hop QA, we have added comprehensive evaluation on **long-document summarization** using the SQuALITY benchmark. QAFD-RAG achieves the strongest overall automated metrics across most measures (Table 3, Page 9), demonstrating that query-aware subgraph retrieval generalizes effectively across diverse reasoning requirements—from multi-hop chains to narrative synthesis—while maintaining the training-free property that enables immediate deployment.
>
> ---
>
> In the revision, we incorporated the following:
>
> ### Modified Section 4.1.4:
>
> **Table 2: Performance on Multi-Hop QA (F1, EM).**
>
> | Method | HotpotQA | 2WikiMultihopQA |
> |--------|----------|-----------------|
> | GraphRAG | (33.4, 14.0) | (15.2, 7.0) |
> | LightRAG | (8.8, 0.0) | (8.2, 1.0) |
> | RAPTOR | (52.3, 25.0) | (38.8, 12.0) |
> | HippoRAG | (58.5, 45.1) | (62.0, 49.5) |
> | QAFD-RAG | **(68.6, 53.7)** | **(62.3, 54.1)** |
>
> ### Added to Section 4.1.3:
>
> **Table 3: Comparison of QAFD-RAG and Baseline Methods on SQuALITY.**
>
> | Method | BLEU-1 | BLEU-2 | ROUGE-1 F1 | ROUGE-2 F1 | METEOR |
> |--------|--------|--------|------------|------------|---------|
> | GraphRAG | 33.91 | 16.12 | 26.38 | 4.08 | 24.38 |
> | HippoRAG | 33.22 | 16.74 | 27.29 | 3.92 | 23.41 |
> | RAPTOR | 32.10 | 16.58 | 25.13 | 3.49 | 22.87 |
> | LightRAG | 34.17 | 17.41 | **28.59** | 4.31 | 23.27 |
> | QAFD-RAG | **35.44** | **18.63** | 28.43 | **4.79** | **25.59** |
>
> ### Added to Section 4.1.3:
>
> *Long-Document Summarization.* We evaluate on SQuALITY (Wang et al., 2022), a long-context QA benchmark where answers are question-focused abstractive summaries of narrative texts. Each question has multiple human-written references, enabling robust evaluation of long-form generation. We report BLEU-1, BLEU-2, ROUGE-1-F1, ROUGE-2-F1, and METEOR against all human references. Full details appear in Appendix A2.1.5.
>
> ### Added to Section 4.1.4:
>
> On the SQuALITY benchmark, QAFD-RAG achieves the highest scores on most automated metrics (BLEU-1, BLEU-2, ROUGE-2 F1, METEOR), showing more faithful and coherent long-form summaries than GraphRAG, HippoRAG, RAPTOR, and LightRAG. Although LightRAG leads on ROUGE-1 F1, QAFD-RAG's query-aligned retrieval provides a stronger overall metric profile, balancing precision and coverage more effectively. On HotpotQA and 2WikiMultiHopQA, QAFD-RAG obtains the highest F1 and EM scores on both benchmarks (Table 2). These improvements in F1 and EM highlight that query-aware diffusion more reliably recovers gold evidence under strict matching. GraphRAG and LightRAG can benefit from broader neighborhood search, but this often hurts precision and consistency. In contrast, QAFD-RAG's query-aligned pruning produces tighter evidence chains and a stable precision–recall balance, enabling superior performance across multi-hop reasoning tasks.
>
> ### Added to Appendix A2.1.5:
>
> We evaluate on all 250 questions from the SQuALITY training dataset (Wang et al., 2022), which contains narrative passages paired with comprehension questions and multiple human-written reference answers per question.
>
> We compute BLEU-1 and BLEU-2 using unigram and bigram modified precisions with geometric averaging. BLEU-n is defined as
>
> $$\text{BLEU-}n := \exp \left( \sum_{i=1}^{n} w_i \log p_i \right),$$
>
> where $p_i$ is the $i$-gram precision and $(w_1,\ldots,w_n)$ are the weights. For BLEU-1 we use $(1,0,0,0)$; for BLEU-2 we use $(0.5,0.5,0,0)$. ROUGE-1 F1 and ROUGE-2 F1 report F1 overlap of unigrams and bigrams using Porter stemming.   METEOR combines unigram precision (P) and recall (R) with a fragmentation penalty:
> $$\text{METEOR} := F_{\text{mean}} \cdot \left(1 - \gamma \cdot \left(\frac{\text{chunks}}{\text{matches}}\right)^\beta\right),$$
> where harmonic mean $F_{\text{mean}} := P \cdot R/(\alpha P + (1-\alpha)R)$, chunks denotes contiguous matched segments, and matches denotes total matched unigrams. We use Natural Language Toolkit (NLTK) defaults: $\alpha=0.9$, $\beta=3.0$, and $\gamma=0.5$. All metrics use lowercased, word-tokenized text with BLEU/METEOR computed against all references.
>
> A knowledge graph is constructed once from all narrative documents and reused across queries. All experiments use hybrid retrieval mode with diffusion parameter a 40-node source budget, and minimum flow threshold $0.01$. Answers are produced with GPT-4o-mini.

---

### Meta-Review · Area_Chair_LQ2p · 2026-01-16

**Summary:**

This work provides a graph based RAG proposal, which incorporates query semantics during graph diffusion process. In indexing stage, they build a KG  and during query stage they dynamically reweights edges and diffuses flow according to query alignment leading to adaptive graph retrieval. The authors argue that this captures query's holistic semantics.  The algorithms is quite scalable. Moreover, the authors experiment with a large number of datasets from Agriculture to Biology domains and show efficacy of the work.

Theoretical analysis of the RAG systems is quite challenging. But I think the paper makes a solid attempt to justify their system in theoretical manner.  Reviewers are generally positive and praised the novelty. One reviewer was negative, but commented on adding more baselines along with few other criticisms. They were clearly addressed by the authors.

Hence I recommend acceptance.

**Reviewer Concerns:**

Most reviewer concerns were addressed by the rebuttal.

**Reviewer Scores:**

Reviewer 6VAy expressed several concerns but I think the authors took care of that. Hence, I recommend acceptance.

---

### Decision · Program_Chairs · 2026-01-26

Accept (Poster)